# A mosaic-type trimeric RBD-based COVID-19 vaccine candidate induces potent neutralization against Omicron and other SARS-CoV-2 variants

Jing Zhang[1,2†], Zi Bo Han[1,2†], Yu Liang[1,2], Xue Feng Zhang[1,2], Yu Qin Jin[1], Li Fang Du[1,2], Shuai Shao[1,2], Hui Wang[3], Jun Wei Hou[1,2], Ke Xu[4], Wenwen Lei[4], Ze Hua Lei[1,2], Zhao Ming Liu[1,2], Jin Zhang[3], Ya Nan Hou[1,2], Ning Liu[1,2], Fu Jie Shen[1,2], Jin Juan Wu[1,2], Xiang Zheng[1,2], Xin Yu Li[1,2], Xin Li[1,2], Wei Jin Huang[5*], Gui Zhen Wu[4*], Ji Guo Su[1,2*], Qi Ming Li[1,2*]

[1]The Sixth Laboratory, National Vaccine and Serum Institute (NVSI), Beijing, China; [2]National Engineering Center for New Vaccine Research, Beijing, China; [3]Beijing Institute of Biological Products Company Limited, Beijing, China; [4]National Institute for Viral Disease Control and Prevention, Chinese Center for Disease Control and Prevention (China CDC), Beijing, China; [5]National Institutes for Food and Drug Control (NIFDC), Beijing, China

*For correspondence:
huangweijin@nifdc.org.cn (WJH);
wugz@ivdc.chinacdc.cn (GZW);
jiguosu@hotmail.com (JGS);
liqiming189@163.com (QML)

†These authors contributed equally to this work

**Abstract** Large-scale populations in the world have been vaccinated with COVID-19 vaccines, however, breakthrough infections of SARS-CoV-2 are still growing rapidly due to the emergence of immune-evasive variants, especially Omicron. It is urgent to develop effective broad-spectrum vaccines to better control the pandemic of these variants. Here, we present a mosaic-type trimeric form of spike receptor-binding domain (mos-tri-RBD) as a broad-spectrum vaccine candidate, which carries the key mutations from Omicron and other circulating variants. Tests in rats showed that the designed mos-tri-RBD, whether used alone or as a booster shot, elicited potent cross-neutralizing antibodies against not only Omicron but also other immune-evasive variants. Neutralizing antibody ID50 titers induced by mos-tri-RBD were substantially higher than those elicited by homo-tri-RBD (containing homologous RBDs from prototype strain) or the BIBP inactivated COVID-19 vaccine (BBIBP-CorV). Our study indicates that mos-tri-RBD is highly immunogenic, which may serve as a broad-spectrum vaccine candidate in combating SARS-CoV-2 variants including Omicron.

## Editor's evaluation

In this work, the authors test, in an animal model, a vaccine booster incorporating three linked SARS-CoV-2 spike receptor binding domain sub-units containing mutations from Omicron sub-variant BA.1 as well as other variants. They demonstrate that this is more effective at boosting neutralizing immunity against Omicron sub-variants and other variants including Beta and Delta than the same vaccine design but incorporating ancestral SARS-CoV-2 spike. While vaccine manufacturers are currently racing to make boosters based on Omicron sub-variant sequences, the approach presented in this paper, which combines mutations in addition to those found on individual Omicron sub-variant sequences, may offer another perspective on how to boost previous vaccine immunity to tackle emerging variants.

**eLife digest** The severe acute respiratory syndrome coronavirus 2 (SARS-CoV-2) pandemic continues to pose a serious threat to public health and has so far resulted in over six million deaths worldwide. Mass vaccination programs have reduced the risk of serious illness and death in many people, but the virus continues to persist and circulate in communities across the globe. Furthermore, the current vaccines may be less effective against the new variants of the virus, such as Omicron and Delta, which are continually emerging and evolving. Therefore, it is urgent to develop effective vaccines that can provide broad protection against existing and future forms of SARS-CoV-2.

There are several different types of SARS-CoV-2 vaccine, but they all work in a similar way. They contain molecules that induce immune responses in individuals to help the body recognize and more effectively fight SARS-CoV-2 if they happen to encounter it in the future. These immune responses may be so specific that new variants of a virus may not be recognized by them. Therefore, a commonly used strategy for producing vaccines with broad protection is to make multiple vaccines that each targets different variants and then mix them together before administering to patients.

Here, Zhang et al. took a different approach by designing a new vaccine candidate against SARS-CoV2 that contained three different versions of part of a SARS-CoV2 protein – the so-called spike protein – all linked together as one molecule. The different versions of the spike protein fragment were designed to include key features of the fragments found in Omicron and several other SARS-CoV-2 variants. The experiments found that this candidate vaccine elicited a much higher immune response against Omicron and other SARS-CoV-2 variants in rats than an existing SARS-CoV-2 vaccine. It was also effective as a booster shot after a first vaccination with the existing SARS-CoV-2 vaccine.

These findings demonstrate that the molecule developed by Zhang et al. induces potent and broad immune responses against different variants of SARS-CoV-2 including Omicron in rats. The next steps following on from this work are to evaluate the safety and immunogenicity of this vaccine candidate in clinical trials. In the future, it may be possible to use a similar approach to develop new broad-spectrum vaccines against other viruses.

## Introduction

The severe acute respiratory syndrome coronavirus 2 (SARS-CoV-2) is continuously evolving, and the emergence of new variants has caused successive waves of coronavirus disease 2019 (COVID-19). Among the circulating variants, five strains, including Alpha, Beta, Gamma, Delta, and Omicron, have been classified into variants of concern (VOCs) by the World Health Organization (WHO) (https://www.who.int/en/activities/tracking-SARS-CoV-2-variants/). Alpha, as the first VOC, became a globally dominant strain in early 2021, which was then replaced by Delta variant from the summer of 2021. These two variants exhibited slightly and moderately less sensitivity to neutralization by serum from vaccinated individuals, with one- to twofold and two- to threefold reductions in neutralizing titers, respectively (*Lipsitch et al., 2022*; *Muik et al., 2021*; *Widge et al., 2021*; *Planas et al., 2021*; *Wu et al., 2021b*; *Pegu et al., 2021*). Beta and Gamma variants outbroke in Africa and South America from early to mid-2021, respectively. Beta variant showed significantly greater immune escape capability and was 3- to 15-fold less susceptible to neutralization by vaccine-induced antibodies (*Lipsitch et al., 2022*; *Planas et al., 2021*; *Edara et al., 2021*; *Shen et al., 2021*). Gamma strain also exhibited obvious reduced neutralizing sensitivity, but the reduction in neutralizing titers was not as substantial as that of Beta strain (*Lazarevic et al., 2021*; *Wang et al., 2021*; *Dejnirattisai et al., 2021*). Besides that, the variants of interest (VOIs) designated by WHO, including Lambda and Mu, have also been reported to have certain immune evasion abilities (*Liu et al., 2021*; *Lou et al., 2021*). The potential immune escape of SARS-CoV-2 variants raised concerns about the efficacy of current COVID-19 vaccines, and new generation vaccines specific to Beta variant have been developed by several groups (*Wu et al., 2021a*; *Callaway and Ledford, 2021*; *Logue et al., 2021*).

Recently, the pandemic of Omicron variant posed a more serious threat to the protective effectiveness of the currently used vaccines. The Omicron variant, also known as B.1.1.529, was first detected in Botswana and reported from South Africa, which was considered to be associated with the sharp rise of infection cases in multiple provinces in South Africa (https://www.who.int/news/

item/26-11-2021-classification-of-omicron-(b.1.1.529)-sars-cov-2-variant-of-concern). Preliminary evidence indicates that this variant may be more transmissible and may have a higher reinfection risk than other VOCs, and thus just two days after its discovery, Omicron has been assigned as a VOC by the WHO (*Vaughan, 2021*; *Callaway, 2021*; *Kupferschmidt and Vogel, 2021*; *Torjesen, 2021*). According to the data from GISGAID, so far, Omicron has spread rapidly to more than 95 countries and the reported cases of this variant are growing rapidly around the world (https://www.gisaid.org/hcov19-variants/). Omicron carries 15 mutations in the receptor binding domain (RBD) of the spike (S) protein that is the immunodominant target of neutralizing antibodies. Compared with other VOCs, the substantially more mutations, as well as their important locations for antibody binding, may enable Omicron to escape the immune protection offered by previous infection or vaccination. Our recent study on the neutralizing sensitivity of the convalescent serum against pseudo-typed Omicron showed that the mean neutralizing titer at a 50% inhibitory dilution (ID50) was significantly decreased 8.4-fold compared to the D614G strain (*Zhang et al., 2022*). Several studies reported by other groups also demonstrated remarkable resistance of Omicron to neutralization by sera from convalescent patients or vaccinated individuals. Neutralizing antibody ID50 titers against Omicron variant in the individuals administered by mRNA COVID-19 vaccines were dramatically reduced 8.6–22 folds compared to the D614G reference strain and by contrast, only 4.3- to 5-fold decline was observed for Beta variant (*Liu et al., 2022*; *Cele et al., 2022*). Sera from the individuals vaccinated with two doses of ChAdOx1 or Ad26.COV2.S failed to neutralize Omicron (*Liu et al., 2022*; *Rössler et al., 2021*). Tests on a panel of existing SARS-CoV-2 neutralizing antibodies showed that Omicron variant evaded neutralization of the majority of these antibodies (*Cao et al., 2022*). As of February 2022, the Omicron variant has evolved into multiple lineages, including BA.1, BA.2, and BA.3. The extensive immune-escape capability of Omicron and other circulating SARS-CoV-2 variants from previous infections and vaccinations raises an urgent need of developing effective broad-spectrum vaccines against these immune-evasive variants.

Guided by structural and computational analyses of S protein RBD, we have developed a trimeric form of RBD vaccine candidate, that is, the homologous trimeric RBD (homo-tri-RBD), in which three RBDs from the prototype SARS-CoV-2 strain were connected end-to-end into a single molecule and co-assembled into a trimeric form (*Liang et al., 2022*). Animal experiments and clinical trials have demonstrated potent protection offered by this vaccine against SARS-CoV-2 (*Liang et al., 2022*; *Kaabi et al., 2022*). At present, homo-tri-RBD has completed phase I/II clinical trial and approved by the United Arab Emirates for emergency use. Here, our vaccine design scheme was extended to broaden its immune response against SARS-CoV-2 variants. Targeting SARS-CoV-2 Omicron and other immune-evasive variants, we present a mosaic-type trimeric RBD (mos-tri-RBD) vaccine candidate, in which the key mutations derived from Omicron (BA.1) as well as other VOCs and VOIs were integrated into the immunogen. The immunogenicity of the designed mos-tri-RBD was evaluated in rats by using live-virus neutralization assays. To illustrate its superiority in stimulating broad-spectrum neutralizing activities against Omicron and other immune-evasive variants, the cross-reactive immunity induced by mos-tri-RBD was compared with that elicited by homo-tri-RBD and the BIBP inactivated COVID-19 vaccine (BBIBP-CorV). Especially, given that large-scale populations worldwide have received the primary series of vaccination, the immunogenicity of the designed mos-tri-RBD as a booster dose following the primary vaccination of BBIBP-CorV was also evaluated and compared with the booster vaccinations of homo-tri-RBD and BBIBP-CorV. Our results showed that the immunization with mos-tri-RBD either alone or as a booster dose elicited potent broad-reactive neutralizing response against SARS-CoV-2 variants including Omicron, which was immunogenically superior to homo-tri-RBD and BBIBP-CorV.

## Results
### Design of the mos-tri-RBD based on the RBDs from SARS-CoV-2 Omicron and other variants

RBD forms a relatively compacted and isolated domain in the structure of spike (S) protein, and the beta-sheets in the core as well as the existence of four disulfide bonds stabilizes the tertiary structure of the domain. The N- and C-termini of RBD are close together, and there exist long loops in both termini. These structural features inspired our construction of a trimeric form of RBD (tri-RBD)

**Table 1.** The details on the eight selected mutations integrated into the two artificially designed RBDs.

| Mutations | Rank of the observed frequency | VOCs and VOIs carrying the mutations |
|---|---|---|
| L452R | 1 | Delta |
| T478K | 2 | Delta, Omicron |
| N501Y | 3 | Alpha, Beta, Gamma, Omicron, Mu |
| E484K | 4 | Beta, Gamma, Mu |
| K417T | 5 | Gamma |
| S477N | 6 | Omicron |
| K417N | 8 | Beta, Omicron |
| F490S | 10 | Lambda |

through an end-to-end connection of three RBDs into a single chain, in which their own long loops at the N- and C-termini serve as the linkers. The designed tri-RBD enables the accommodation of three RBDs in one immunogen, which can be extended to include Omicron RBD into the immunogen.

In this study, mos-tri-RBD was designed targeting Omicron as well as other emerging variants with distinct immune evasion capability. Mos-tri-RBD also consisted of three RBDs, one of which was derived from Omicron and the other two were artificially designed to carry the key mutations appearing in SARS-CoV-2 variants. These key mutations integrated into mos-tri-RBD were selected as those appearing in SARS-CoV-2 VOCs or VOIs and simultaneously being ranked in the top ten most frequently occurring mutations in RBD as counted by Wei group (https://weilab.math.msu.edu/MutationAnalyzer/). According to this criterion, a total of eight mutations were chosen and introduced into the two artificially designed RBDs (The details on these mutations were provided in *Table 1*), where one RBD contained the mutations of K417N, L452R, T478K, F490S and N501Y, and the other included K417T, S477N and E484K (*Figure 1A*). Many pieces of evidence have indicated that these mutations largely contribute to the immune escape of the related variants. We sought to integrate these key mutations into a single immunogen to elicit cross-neutralization against not only SARS-CoV-2 Omicron but also other circulating variants. To facilitate the self-trimerization of mos-tri-RBD, for each RBD the residues 319–537 were truncated from the S protein to retain the long loops at both termini, as shown in *Figure 1A*. Our previous studies have shown that the RBDs with this truncation scheme can correctly fold and co-assemble into a trimeric structure (*Liang et al., 2022*).

## Expression, identification, and characterization of the recombinant mos-tri-RBD protein

The recombinant mos-tri-RBD was expressed using CHO cells, and then purified by chromatography and ultrafiltration as described previously (*Liang et al., 2022*). Based on the sequence of the designed mos-tri-RBD, the theoretically calculated mass of each RBD was about 24–25 kDa, and that of the entire mos-tri-RBD was approximately 74 kDa. SDS-PAGE analysis displayed an obvious band corresponding to the molecular mass of about 92 kDa (*Figure 1B*), implying the formation of the trimeric RBD. The measured mass was larger than the theoretical value calculated by the sequence, which was attributed to the heavy glycosylation of the protein as discussed in our previous studies (*Liang et al., 2022*). To determine the biological function of the recombinant mos-tri-RBD, its binding ability to an RBD-specific monoclonal neutralizing antibody MM117 was tested using enzyme-linked immunosorbent assay (ELISA). MM117 has been proved to be able to bind specifically with the RBDs of SARS-CoV-2 prototype, Omicron and Delta strains. Protein concentration-dependent binding activity was observed for MM117, suggesting the formation of native conformation of the RBDs in mos-tri-RBD (*Figure 1C*). Furthermore, the binding avidity of mos-tri-RBD with the receptor human angiotensin converting enzyme 2 (hACE2) was also measured using surface plasmon resonance (SPR) assay. The association rate constant ($k_a$) and dissociation rate constant ($k_d$) were quantified to be $3.87 \times 10^7\ M^{-1}s^{-1}$ and $2.59 \times 10^{-4}\ s^{-1}$, respectively, and thus the apparent dissociation constant $K_D$ was determined to be $6.69 \times 10^{-3}\ nM$ (*Figure 1D*). Many studies have revealed that the dissociation constant for the prototype monomeric RBD binding to hACE2 was in the range of $2.66 - 26.34\ nM$, and those for the Beta and Omicron monomeric RBDs were $1.75 - 13.83\ nM$ and $2.48 - 31.40\ nM$, respectively (*Routhu et al., 2021*; *Laffeber et al., 2021*; *Xu et al., 2022a*; *Xu et al., 2022b*; *Lan et al., 2022*; *Han et al., 2022*). The studies of Routhu et al. demonstrated that the hACE2 binding avidity of the RBD trimer was about 500-fold higher than that of the RBD monomer (*Routhu et al., 2021*). Our previous studies showed

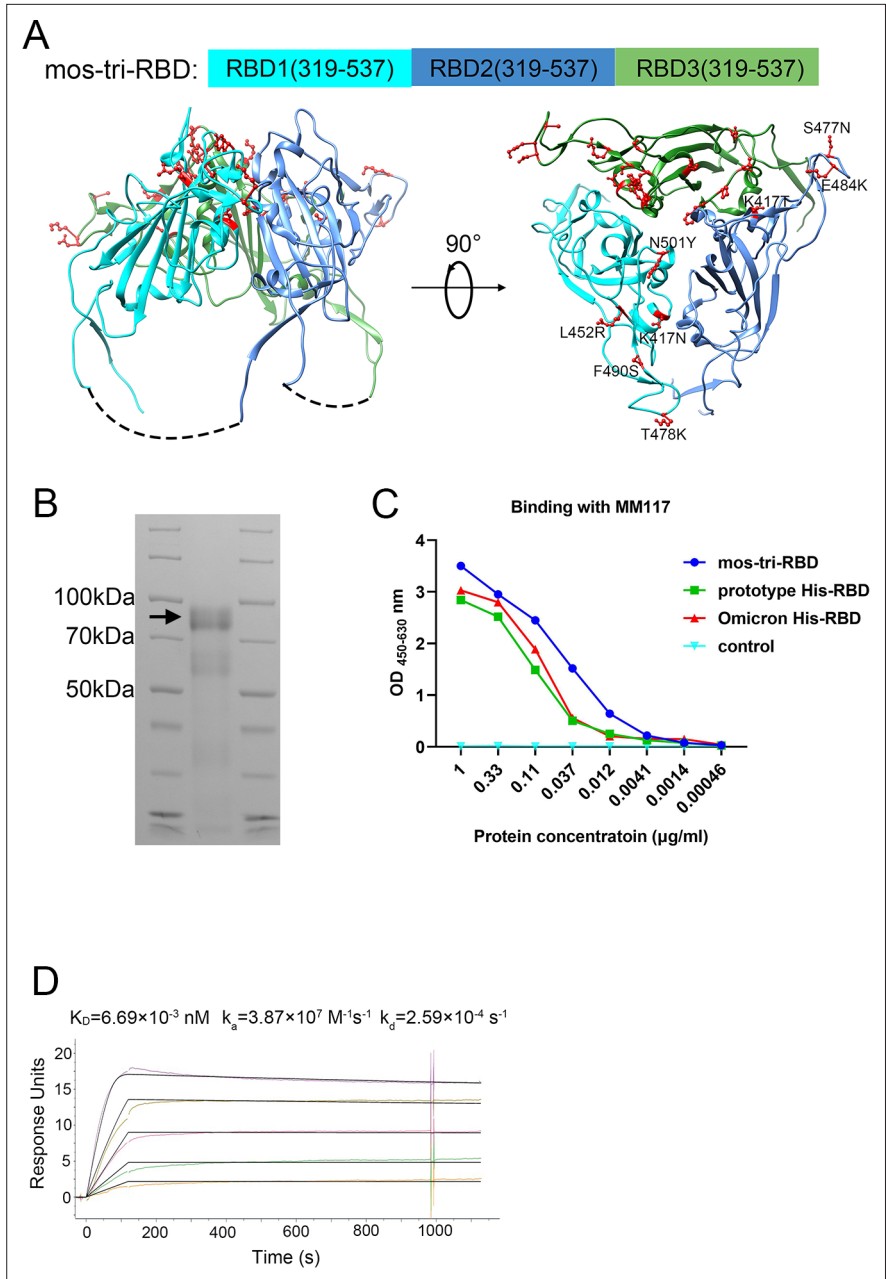

**Figure 1.** Design, expression and characterization of the mosaic-type trimeric form of RBD (mos-tri-RBD). (**A**) Schematic illustration of the designed mos-tri-RBD. In mos-tri-RBD, three heterologous RBDs were connected end to end into a single chain and co-assembled into a trimeric structure. For the three RBDs, one was derived from the Omicron (BA.1) variant (green color), and the other two were artificially designed harboring the key immune-evasion-related mutations that emerged in SARS-CoV-2 variants, in which one contained the mutations of K417N, L452R, T478K, F490S, and N501Y (cyan color), and the other one contained K417T, S477N, and E484K (blue color). These mutations are highlighted in the red ball-and-stick model in the figure. Each RBD subunit in mos-tri-RBD was composed of the residues 319–537 from the spike protein. The dotted curves in the figure represent the direct connection between the C-terminus of the former RBD and the N-terminus of the latter RBD. The schematic structure of mos-tri-RBD was drawn by Chimera software (*Pettersen et al., 2004*) based on the PDB file with accession number 6zgi. (**B**) SDS-PAGE analysis of the recombinant mos-tri-RBD. (**C**) Concentration-dependent binding ability of mos-tri-RBD with an RBD-specific monoclonal neutralizing antibody MM117 tested using ELISA. (**D**) Binding avidity of mos-tri-RBD with the receptor hACE2 measured using SPR assay. In this figure, different curves represent different concentrations of analyte (top to bottom: 263.70 ng/ml, 131.85 ng/ml, 65.93 ng/ml, 32.96 ng/ml, and 16.48 ng/ml). Both the original (color curves) and fitted (black curves) data are displayed.

*Figure 1 continued on next page*

*Figure 1 continued*

The online version of this article includes the following source data for figure 1:

**Source data 1.** The raw files of SDS-PAGE results.

**Source data 2.** Concentration-dependent binding ability of mos-tri-RBD with the antibody MM117.

that the apparent dissociation constant for another designed trimeric RBD protein, which is composed of three RBDs derived respectively from the prototype, Beta and Kappa viruses, binding to hACE2 was $3.20 \times 10^{-3}$ *nM* (*Liang et al., 2022*). The SPR detection results for the designed mos-tri-RBD were consistent with our previous studies (*Liang et al., 2022*) and the results reported by Routhu et al. (*Routhu et al., 2021*). The results of SPR assay verified the functionality of the designed mos-tri-RBD and the correct folding of each RBD into its native conformation. All these results suggested that the designed mos-tri-RBD assembled into a trimeric form and each RBD subunit correctly folded into its native structure.

## Mos-tri-RBD induced potent cross-reactive neutralizing response against the live viruses of SARS-CoV-2 Omicron and other immune-evasive variants

To evaluate the cross-reactive immunogenicity of the designed mos-tri-RBD, we intramuscularly immunized rats using two doses of mos-tri-RBD mixed with Aluminum adjuvant three weeks apart. Another three groups of rats received two doses of homo-tri-RBD, BBIBP-CorV or adjuvant, respectively, with the same immunization regimen were used for comparison. Sera from the immunized rats were collected on day 7 after the last vaccination (*Figure 2A*). Neutralizing ID50 titers in the sera against multiple SARS-CoV-2 strains, including prototype, Omicron (BA.1.1), Beta and Delta strains, were detected using live-virus neutralization assay.

Live-virus neutralization assay showed that compared with the prototype virus, the Omicron variant exhibits substantially less susceptibility to neutralization elicited by two doses of BBIBP-CorV vaccination. The geometric mean titer (GMT) of neutralizing antibodies against prototype strain was 699, whereas the Omicron-specific neutralizing ID50 in half of the rats was less than the detectable limit of the assay (*Figure 2B*), suggesting substantial evasion of Omicron variant from the immunity elicited by BBIBP-CorV. The result was consistent with the findings of other studies that Omicron exhibited significant immune escape capability (*Zhang et al., 2022*; *Liu et al., 2022*; *Cele et al., 2022*; *Rössler et al., 2021*; *Cao et al., 2022*). Compared with BBIBP-CorV, homo-tri-RBD vaccination significantly improved the neutralizing antibody GMT against Omicron virus from <31 to 1077, with a more than 34.7-fold increase. Furthermore, remarkably enhanced neutralizing ID50 titers against Omicron were elicited by mos-tri-RBD vaccination, in which the neutralizing GMT reached 2876, with 2.7-fold and >92.8-fold increases in comparison to the homo-tri-RBD and BBIBP-CorV vaccinations, respectively (*Figure 2B*). Statistical analysis showed that the anti-Omicron neutralizing antibody response elicited by mos-tri-RBD was significantly higher than that of homo-tri-RBD (p=0.0057) and the neutralization elicited by homo-tri-RBD was also significantly higher than that of BBIBP-CorV (p<0.0001). Our study demonstrated that the designed mos-tri-RBD exhibited much higher immunogenicity against Omicron variant than homo-tri-RBD and BBIBP-CorV. Mos-tri-RBD may serve as an effective vaccine candidate in fighting against Omicron variant.

Similar results were also observed for Beta and Delta variants. In the rats immunized with BBIBP-CorV, the neutralizing antibody GMT against Beta variant was reduced by 2.1-fold in comparison with that against the prototype virus, suggesting a considerable immune escape of the variant. Many previously reported results also found that Beta variant distinctly evaded the immunity offered by natural infection or vaccination (*Lipsitch et al., 2022*; *Planas et al., 2021*; *Edara et al., 2021*; *Shen et al., 2021*). While, compared to BBIBP-CorV, the anti-Beta neutralizing antibody GMT elicited by homo-tri-RBD was increased from 328 to 2547 (7.8-fold), and further improved to 6917 (21.1-fold) by mos-tri-RBD vaccination (*Figure 2B*). Similarly, for Delta variant, the neutralizing antibody GMTs induced by homo-tri-RBD and mos-tri-RBD were 5.1-fold and 8.6-fold, respectively, higher than that induced by BBIBP-CorV (*Figure 2B*). Our results indicated that besides Omicron variant, mos-tri-RBD was also highly immunogenic against Beta and Delta variants.

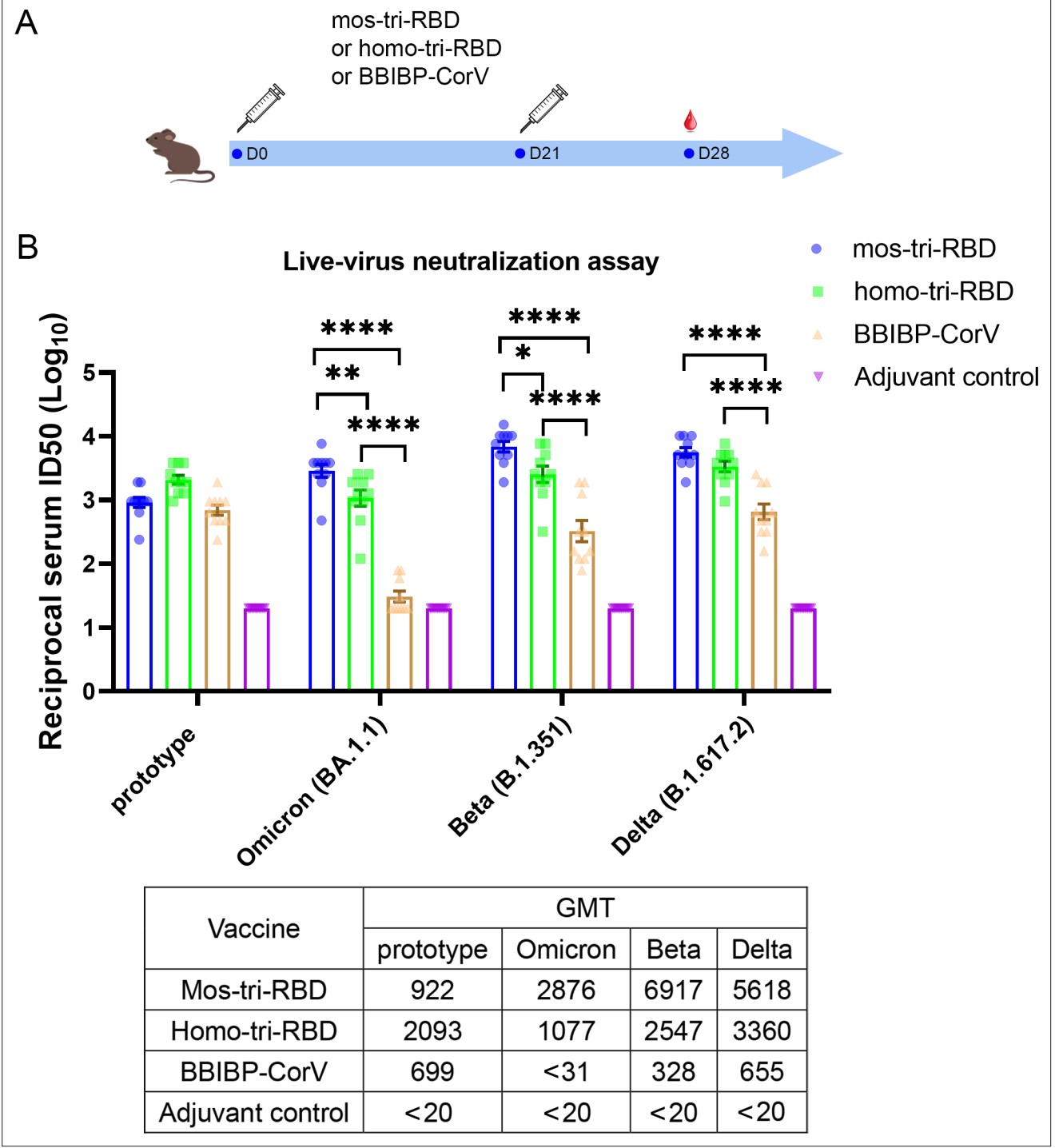

**Figure 2.** Evaluation of the cross-reactive immunogenicity of mos-tri-RBD against multiple SARS-CoV-2 strains, including prototype, Omicron, Beta and Delta strains, using live-virus neutralization assay. (**A**) Timeline of rat immunization and serum collections. A group of Wistar rats (n=10 with half male and half female) were immunized intramuscularly with two doses of mos-tri-RBD with three weeks apart. Another three groups of rats received two doses of homo-tri-RBD, BBIBP-CorV and adjuvant, respectively, were used for comparison (n=10 rats per group with half male and half female). Sera from all the immunized rats were collected on day 7 after the last vaccination. (**B**) The reciprocal neutralizing ID50 titers in the sera elicited by mos-tri-RBD compared with those elicited by homo-tri-RBD and BBIBP-CorV against the live-viruses of SARS-CoV-2 prototype strain, and Omicron, Beta, and Delta variants. The quantification limit of the live-virus neutralization assay was 20, and the ID50 titers below the limit of quantification (LOQ) were set to 20. Data are presented as mean ± SEM. One-way ANOVA followed by the LSD t-test was used for the comparison of data between different groups. *p<0.05, **p<0.01, ****p<0.0001. GMT values are displayed in the lower part of the figure.

*Figure 2 continued on next page*

*Figure 2 continued*

The online version of this article includes the following source data for figure 2:

**Source data 1.** Individual data of live-virus neutralizing ID50 titers against several SARS-CoV-2 circulating strains in the sera elicited by mos-tri-RBD compared with those elicited by homo-tri-RBD and BBIBP-CorV.

In summary, live-virus neutralization assays demonstrated that the designed mos-tri-RBD, which integrated key residues from Omicron and other circulating SARS-CoV-2 variants into a single antigen, could serve as a broad-spectrum COVID-19 vaccine candidate against not only Omicron variant but also other SARS-CoV-2 variants. However, it should be noted that due to the absence of wild-type RBD in the mosaic antigen, the neutralizing antibody response against SARS-CoV-2 prototype strain stimulated by mos-tri-RBD was lower than that by homo-tri-RBD, but still comparable to that by BBIBP-CorV (*Figure 2B*).

## Mos-tri-RBD as a booster dose induced cross-neutralization against the pseudo-typed SARS-CoV-2 Omicron as well as other VOCs and VOIs

Given that large-scale populations worldwide have received the primary series of vaccination, the immunogenicity of the designed mos-tri-RBD as a booster dose was mainly evaluated in this study. Rats were primed with a dose of BBIBP-CorV, and successively boosted with a dose of mos-tri-RBD ('BBIBP-CorV +mos-tri-RBD' group), homo-tri-RBD ('BBIBP-CorV +homo-tri-RBD' group), or BBIBP-CorV ('BBIBP-CorV +BBIBP-CorV' group), as shown in *Figure 3A*. Another group of rats received two doses of adjuvant was served as a control. On day 7 post-boost, the sera from the immunized rats were collected, and the neutralizing response against various SARS-CoV-2 strains was tested using pseudo-virus neutralization assays.

Pseudo-virus neutralization assay demonstrated that all the three prime-boosting vaccinations induced elevated neutralizing antibodies in comparison to the adjuvant control group against the pseudo-virus of SARS-CoV-2 prototype strain. Although mos-tri-RBD did not contain the prototype RBD, the neutralizing ID50 titers against prototype strain in the sera elicited by 'BBIBP-CorV +mos-tri-RBD' were no less than those elicited by 'BBIBP-CorV +homo-tri-RBD', both of which were significantly higher than those induced by 'BBIBP-CorV +BBIBP-CorV' vaccinations (*Figure 3B*). Our results indicated that mos-tri-RBD was highly immunogenic as a booster dose against the prototype SARS-CoV-2 strain.

Compared with the prototype pseudo-virus, the neutralizing antibody GMTs against Omicron variant, including lineages BA.1, BA.2, and BA.3, elicited by 'BBIBP-CorV +BBIBP-CorV' vaccination were significantly reduced (*Figure 3B*), suggesting high immune escape capability of Omicron variant from the immunity offered by BBIBP-CorV. In comparison to homologous booster of BBIBP-CorV, the 'BBIBP-CorV +homo-tri-RBD' vaccination improved the neutralizing antibody GMTs against Omicron BA.1, BA.2, and BA.3 pseudo-viruses from 383,<175 and <100 to 932, <372 and<340, respectively, but these values were also significantly lower than the value against the prototype strain (*Figure 3B*). Furthermore, remarkably enhanced neutralizing ID50 titers against Omicron were elicited by 'BBIBP-CorV +mos-tri-RBD' vaccination, in which the neutralizing GMTs reached 3249, 1459, and 957 against BA.1, BA.2, and BA.3, respectively. The neutralizing GMTs boosted by mos-tri-RBD against the three lineages of Omicron variant were 3.5-fold, >3.9-fold, and >2.8-fold higher than those boosted by homo-tri-RBD, and 8.5-fold, >8.3-fold, and >9.6-fold higher than by BBIBP-CorV, respectively (*Figure 3B*). Mos-tri-RBD was immunogenically superior to homo-tri-RBD and BBIBP-CorV as a booster vaccine for BBIBP-CorV recipients against Omicron variant.

Considering that mos-tri-RBD also contains the key mutations from other SARS-CoV-2 variants with potential immune evasion ability, we then evaluate whether the mos-tri-RBD booster induced higher neutralizing responses than homo-tri-RBD and BBIBP-CorV against other pseudo-typed SARS-CoV-2 VOCs and VOIs, including Alpha, Beta, Delta, Gamma, Lambda, and Mu variants. Pseudo-virus neutralization assays showed that for most of the tested variants, the neutralizing ID50 titers in the sera elicited by 'BBIBP-CorV +mos-tri-RBD' vaccination were higher than those by 'BBIBP-CorV +mos-tri-RBD' and 'BBIBP-CorV +BBIBP-CorV'. Especially, for Beta, Gamma, Delta, Lambda, and Mu variants, the neutralizing antibody GMTs were increased 8.1-fold, 5.1-fold, 9.1-fold, 9.1-fold, and 5.6-fold, respectively, for 'BBIBP-CorV +mos-tri-RBD' vaccination compared to 'BBIBP-CorV +BBIBP-CorV'

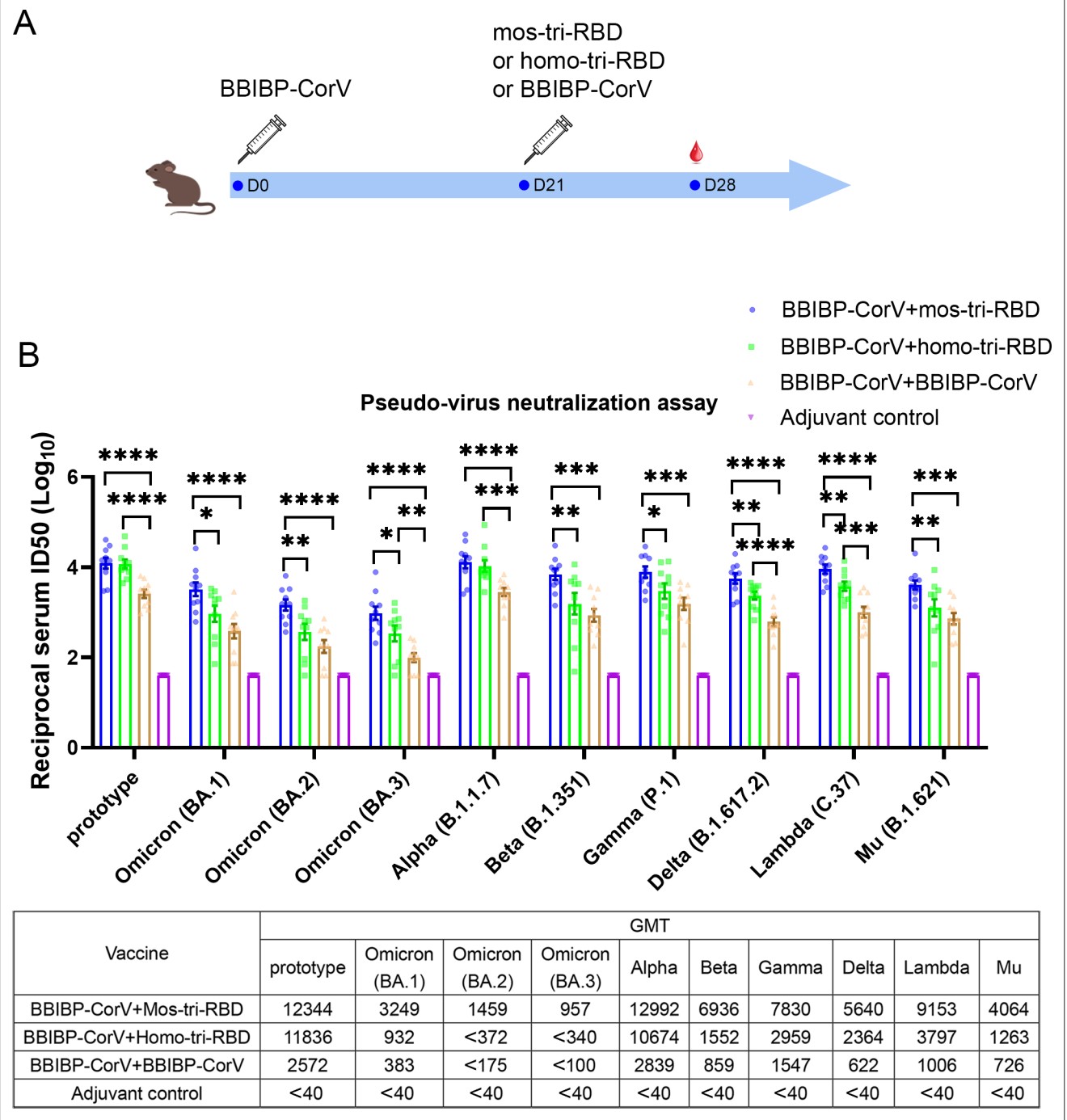

| Vaccine | GMT | | | | | | | | | |
|---|---|---|---|---|---|---|---|---|---|---|
| | prototype | Omicron (BA.1) | Omicron (BA.2) | Omicron (BA.3) | Alpha | Beta | Gamma | Delta | Lambda | Mu |
| BBIBP-CorV+Mos-tri-RBD | 12344 | 3249 | 1459 | 957 | 12992 | 6936 | 7830 | 5640 | 9153 | 4064 |
| BBIBP-CorV+Homo-tri-RBD | 11836 | 932 | <372 | <340 | 10674 | 1552 | 2959 | 2364 | 3797 | 1263 |
| BBIBP-CorV+BBIBP-CorV | 2572 | 383 | <175 | <100 | 2839 | 859 | 1547 | 622 | 1006 | 726 |
| Adjuvant control | <40 | <40 | <40 | <40 | <40 | <40 | <40 | <40 | <40 | <40 |

**Figure 3.** Evaluation of the cross-reactive immunogenicity of mos-tri-RBD as a booster shot against SARS-CoV-2 Omicron as well as other VOCs and VOIs using pseudo-virus neutralization assays. (**A**) Timeline of rat immunization and serum collections. Three groups of Wistar rats (n=10 rats per group with half male and half female) were primed with a dose of BBIBP-CorV and boosted by mos-tri-RBD, homo-tri-RBD or BBIBP-CorV with three weeks apart. Another group of rats (n=10 with half male and half female) vaccinated with two doses of adjuvant served as control. The sera of all the immunized rats were collected on day 7 post-boosting immunization. (**B**) The reciprocal neutralizing ID50 titers in the sera elicited by 'BBIBP-CorV +mos-tri-RBD' vaccination compared with those elicited by 'BBIBP-CorV +homo-tri-RBD' and 'BBIBP-CorV +BBIBP-Corv' vaccinations against the pseudo-viruses of SARS-CoV-2 Omicron as well as other VOCs and VOIs. The quantification limit of the pseudo-virus neutralization assay was 40, and the ID50 titers below the LOQ were set to 40. Data are presented as mean ± SEM. One-way ANOVA followed by the LSD t-test was used for the comparison of data between different groups. *p<0.05, **p<0.01, ****p<0.0001. GMT values are displayed in the lower part of the figure.

The online version of this article includes the following source data for figure 3:

**Source data 1.** Individual data of pseudo-virus neutralizing ID50 titers against various SARS-CoV-2 circulating strains in the sera elicited by 'BBIBP-CorV +mos-tri-RBD' vaccination compared with those elicited by 'BBIBP-CorV +homo-tri-RBD' and 'BBIBP-CorV +BBIBP-CorV' vaccinations.

vaccination, and 4.5-fold, 2.6-fold, 2.4-fold, 2.4-fold, and 3.2-fold, respectively, compared to 'BBIBP-CorV +homo-tri-RBD' vaccination (*Figure 3B*). These results indicated that mos-tri-RBD as a booster dose significantly improved the immunogenicity against not only Omicron variant but also other potentially immune-evasive SARS-CoV-2 variants. Mos-tri-RBD may act as a booster vaccine with broad-neutralization activities.

### Mos-tri-RBD as a booster dose induced cross-neutralization against the live viruses of SARS-CoV-2 prototype, Omicron, Beta and Delta strains

The significantly higher neutralizing activities boosted by mos-tri-RBD against multiple SARS-CoV-2 variants, including Omicron (BA.1.1), Beta and Delta, were further verified by using live virus neutralization assays. As a comparison, the neutralizing response against prototype strain was also tested.

Regarding the prototype strain, live virus neutralization assays showed that booster vaccinations with the three vaccines all elicited strong neutralization activities compared to adjuvant control. In line

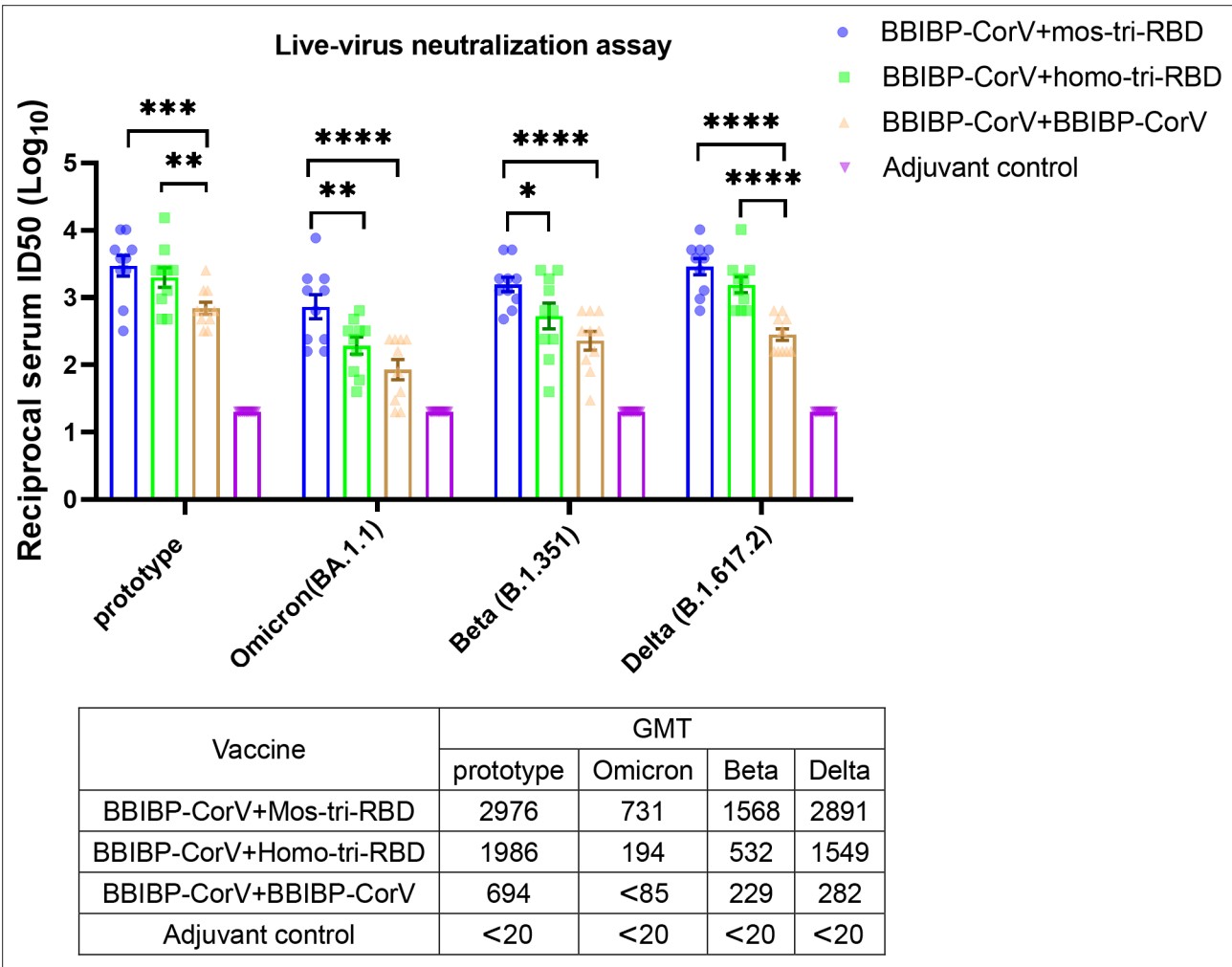

**Figure 4.** Evaluation of the cross-reactive immunogenicity of mos-tri-RBD as a booster shot against multiple SARS-CoV-2 strains, including prototype, Omicron, Beta, and Delta strains, using live-virus neutralization assay. The reciprocal neutralizing ID50 titers in the sera elicited by 'BBIBP-CorV +mos-tri-RBD' vaccination compared with those elicited by 'BBIBP-CorV +homo-tri-RBD' and 'BBIBP-CorV +BBIBP-Corv' vaccinations against the live-viruses of SARS-CoV-2 Omicron as well as other immune-evasive variants. The quantification limit of the live-virus neutralization assay was 20, and the ID50 titers below the LOQ were set to 20. Data are presented as mean ± SEM. One-way ANOVA followed by the LSD t-test was used for the comparison of data between different groups. *p<0.05, **p<0.01, ****p<0.0001. GMT values are displayed in the lower part of the figure.

The online version of this article includes the following source data for figure 4:

**Source data 1.** Individual data of live-virus neutralizing antibody ID50 titers against several SARS-CoV-2 circulating strains in the sera elicited by 'BBIBP-CorV +mos-tri-RBD' vaccination compared with those elicited by 'BBIBP-CorV +homo-tri-RBD' and 'BBIBP-CorV +BBIBP-CorV' vaccinations.

with the results of pseudo-virus neutralization assay, the live-virus neutralizing antibody ID50 titers induced by 'BBIBP-CorV +mos-tri-RBD' and 'BBIBP-CorV +homo-tri-RBD' vaccinations were significantly greater than those by 'BBIBP-CorV +BBIBP-CorV' (*Figure 4*). The results demonstrated that mos-tri-RBD and homo-tri-RBD were more immunogenic than BBIBP-CorV against the prototype SARA-CoV-2 strain.

The neutralizing antibody response against Omicron variant boosted by BBIBP-CorV was dramatically reduced compared with that against prototype virus, suggesting significant immune evasion of Omicron from the homologous BBIBP-CorV booster vaccination. Compared with 'BBIBP-CorV +BBIBP-CorV' vaccination, the neutralizing antibody GMT against Omicron variant in the 'BBIBP-CorV +homo-tri-RBD' immunization group was increased from <85 to 194, however, the value was also significantly lower than that against the prototype strain. Furthermore, similar to the results of pseudo-virus neutralization assay, 'BBIBP-CorV +mos-tri-RBD' vaccination elicited remarkably improved live-virus neutralizing antibodies with a GMT value of 731 (*Figure 4*). Compared to booster vaccinations of BBIBP-CorV and homo-tri-RBD, boosting with mos-tri-RBD induced >8.6-fold and 3.8-fold higher Omicron-specific neutralizing antibodies, respectively, which provides an effective booster vaccine against Omicron variant. Similar results were also observed for Beta and Delta variants. In 'BBIBP-CorV +BBIBP-CorV' vaccination groups, both Beta and Delta variants exhibited less sensitivity to neutralization by the sera from the immunized rats. While, the neutralizing antibody GMT against Beta variant elicited by 'BBIBP-CorV +homo-tri-RBD' was increased from 229 to 532 (2.3-fold), and further improved to 1568 (6.8-fold) by 'BBIBP-CorV +mos-tri-RBD' vaccination (*Figure 4*). For Delta variant, the neutralizing antibody GMT boosted by mos-tri-RBD and homo-tri-RBD was 5.5-fold and 10.3-fold, respectively, higher than that boosted by BBIBP-CorV (*Figure 4*). These results implied that mos-tri-RBD was immunologically superior to homo-tri-RBD and BBIBP-CorV as a booster dose against Omicron and other immune-evasive SARS-CoV-2 variants.

In summary, both pseudo- and live-virus neutralization assays demonstrated that the designed mos-tri-RBD could serve as an effective booster vaccine to elicit potent and cross-reactive immunity against not only prototype virus but also Omicron and other variants.

## Discussion

Several studies indicated that Omicron variant might lead to less severe disease than earlier pandemic variants (*Ledford, 2021*; *Sigal et al., 2022*; *Wolter et al., 2022*), in which the hospitalization risk was reduced by 29% for Omicron compared with Delta variant, and the fatality rates decreased from 3.4% for Delta to 1.9% for Omicron. However, Omicron variant is still a global threat due to its higher transmissibility and increased resistance to antiviral immunity. Facing the severe pandemic of Omicron, WHO has recommended updating the composition of current COVID vaccines and developing multivalent or cross-protective vaccines against the variants (https://www.who.int/news/item/). Here, we designed a mosaic-type tri-RBD (mos-tri-RBD), which harbors the key mutations derived from Omicron and other immune-evasive variants, to broaden the immune response to SARS-CoV-2 variants. The immunogenicity of the designed mosaic-type vaccine candidate was assessed in rats, and live-virus neutralization assays showed that mos-tri-RBD elicited broad-spectrum neutralizing antibody response against multiple SARS-CoV-2 variants including Omicron. Considering that large-scale populations in the world have completed the primary series of vaccination, the immunogenicity of the designed mos-tri-RBD was also evaluated as a booster dose following the vaccination of BBIBP-CorV. Tests in rats showed that the mos-tri-RBD booster vaccination elicited more than 2.8–3.9-fold higher neutralizing antibodies against Omicron compared to the booster vaccination of homo-tri-RBD, and more than 8.3–9.6-fold higher compared to the booster of BBIBP-CorV. For other SARS-CoV-2 VOCs and VOIs, the neutralizing antibody titers induced by the mos-tri-RBD booster were 1.2–4.5-fold higher than the homo-tri-RBD booster, and 4.6–10.3-fold higher than the BBIBP-CorV booster. Thus, mos-tri-RBD may serve as a broad-neutralizing vaccine candidate, which could be used alone or as a booster shot in combating SARS-CoV-2 variants including Omicron.

A commonly used strategy for the development of broad-spectrum vaccines is to produce polyvalent vaccines that contain multiple strain-specific monovalent components. Considering the potential immune escape capability of Beta variant, several studies have designed the Beta-specific COVID-19 vaccines and applied combining with the anti-prototype ones to broaden immune response (*Wu et al., 2021a*; *Callaway and Ledford, 2021*; *Logue et al., 2021*). Targeting the Omicron strain with stronger

immune evasion ability, several vaccine manufacturers have announced the update of the composition of their COVID-19 vaccines to provide effective protection against Omicron (*Cohen, 2021*). In the present work, we provided another strategy for the development of broad-spectrum vaccines against SARS-CoV-2, that is, the construction of mosaic-type vaccines which incorporate multiple antigens and key mutations derived from different variants into a single hybrid immunogen. The mosaic strategy has been successfully applied to the development of broad-spectrum vaccines for HIV, coronaviruses and influenza (*Barouch et al., 2010*; *Cohen et al., 2021*; *Kanekiyo et al., 2019*), and several studies have demonstrated that mosaic-type immunogen elicited superior B cell responses both in quantity and quality compared to the homotypic immunogens (*Kanekiyo et al., 2019*). Our results also showed that the constructed mos-tri-RBD not only strengthens but also broadens neutralizing response against SARS-CoV-2.

SARS-CoV-2 virus continuously evolves, and the mutation rate was estimated to be $1.12 \times 10^{-3}$ mutations per site-year (*Koyama et al., 2020*; *Amicone et al., 2022*). Therefore, it is believed that the virus may acquire new mutations, and new variants will continue to emerge. Our mosaic-type vaccine can be easily modified to incorporate new residue mutations to fight against the possible emerging variants in the future.

The construction of mosaic-type immunogen, which combines the key mutations relevant to immune evasion into a single molecule, provides an effective strategy to achieve broad-spectrum neutralization in a single-component vaccine. The mosaic strategy may also be used in the developments of mRNA- and DNA-based COVID-19 vaccines.

# Materials and methods

## Key resources table

| Reagent type (species) or resource | Designation | Source or reference | Identifiers | Additional information |
|---|---|---|---|---|
| Strain, strain background (*SARS-CoV-2 virus*) | Live SARS-CoV-2 prototype virus (QD-01 strain) | National Institute for Viral Disease Control and Prevention, China CDC | | |
| Strain, strain background (*SARS-CoV-2 virus*) | Live SARS-CoV-2 Omicron virus (NPRC 2.192100005 strain) | National Institute for Viral Disease Control and Prevention, China CDC | | |
| Strain, strain background (*SARS-CoV-2 virus*) | Live SARS-CoV-2 Beta virus (GD84 strain) | National Institute for Viral Disease Control and Prevention, China CDC | | |
| Strain, strain background (*SARS-CoV-2 virus*) | Live SARS-CoV-2 Delta virus (GD96 strain) | National Institute for Viral Disease Control and Prevention, China CDC | | |
| Strain, strain background (*SARS-CoV-2 pseudo-virus*) | SARS-CoV-2 prototype, Omicron (BA.1), Omicron (BA.2), Omicron (BA.3), Alpha, Beta, Delta, Gamma, Lambda and Mu pseudo-viruses | *Wang et al., 2021*; *Li et al., 2021*; *Zhang et al., 2021*; *Nie et al., 2020* | | |
| Cell line (*CHO*) | CHO-K1 cell line | ATCC | Cat#CCL-61, RRID:CVCL_0214 | |
| Cell line (*Homo-sapiens*) | Huh-7 cells | JCRB | Cat#JCRB0403; RRID: CVCL_0336 | |
| Cell line (Chlorocebus sabaeus) | Vero cells | National Institute for Food and Drug Control (NIFDC), Beijing, China | | |
| Biological sample (*Wistar rats*) | Serum samples from immunized Wistar rats | This paper | | Freshly isolated from immunized rats |
| Antibody | Mouse monoclonal anti-RBD | Sino Biological Inc, China | Cat#40592-MM117 | ELISA (1 µg/mL) |
| Antibody | Goat polyclonal anti-mouse IgG-HRP | ZSGB-BIO | Cat#ZB-2305; RRID: AB_2747415 | ELISA (1:10000) |

*Continued on next page*

*Continued*

| Reagent type (species) or resource | Designation | Source or reference | Identifiers | Additional information |
|---|---|---|---|---|
| Recombinant DNA reagent | Plasmid-SARS-CoV-2-mos-tri-RBD | This paper | | Reference to "protein expression and purification" section |
| Peptide, recombinant protein | Recombinant mos-tri-RBD protein (mammalian cell-expressed) | This paper | | Reference to "protein expression and purification" section |
| Peptide, recombinant protein | Recombinant monomeric his-tagged RBD of the prototype SARS-CoV-2 strain (Baculovirus-insect cell-expressed) | Sino Biological Inc, China | Cat#40592-V08B | |
| Peptide, recombinant protein | Recombinant monomeric his-tagged RBD of the Omicron (B.1.1.529) SARS-CoV-2 strain (HEK 293 cell-expressed) | Sino Biological Inc, China | Cat#40592-V08H121 | |
| Peptide, recombinant protein | Recombinant hACE2 protein (mammalian cell-expressed) | Sino Biological Inc, China | Cat#10108-H08H | |
| Commercial assay or kit | SPR | BIAcore 8 k, GE Healthcare | N/A | |
| Chemical compound, drug | Aluminum hydroxide adjuvant | This paper | N/A | Produced by the reaction of aluminum chloride and sodium hydroxide |
| Software, algorithm | UCSF Chimera | Chimera team at University of California | https://www.cgl.ucsf.edu/chimera/ | |
| Software, algorithm | BIAcore Insight Evaluation Software | GE Healthcare | | |
| Software, algorithm | GraphPad Prism version 8 | GraphPad Software | https://www.graphpad.com/ | |

## Cells and viruses

All cell lines tested negative for mycoplasma. CHO-K1 cells were from ATCC (Cat: CCL-61) and tested using isoenzyme analysis method. Huh-7 cells were from the Japanese Collection of Research Bioresources (Cat: JCRB0403) and authenticated using STR profiling method. Huh-7 cells were grown in Dulbecco's modified Eagle medium (DMEM) supplied with 100 U/mL of Penicillin-Streptomycin solution, 20 mM N-2-hydroxyethylpiperazine-N-2-ethane sulfonic acid (HEPES) and 10% fetal bovine serum (FBS), at 37 °C and in a 5% $CO_2$ environment. Vero cells were from National Institute for Food and Drug Control (NIFDC) of China and authenticated using STR profiling method. Vero cells were cultured in Medium 199 containing 5% FBS, at 37 °C and in 5% $CO_2$. The live viruses of SARS-CoV-2 prototype, Omicron, Beta, and Delta strains were isolated by the National Institute for Viral Disease Control and Prevention, Chinese Center for Disease Control and Prevention (China CDC). The SARS-CoV-2 viruses were grown in Vero cells.

## Protein expression and purification

The mosaic-type trimeric RBD (mos-tri-RBD) protein was constructed through connecting three heterologous RBDs (amino acid 319–537 in S proteins) into a single chain, which co-assembled into a trimeric structure. The designed protein was transiently expressed in CHO cells and purified by chromatography combined with ultrafiltration, as described in our previous paper (*Liang et al., 2022*). Briefly, the sequence of the designed mos-tri-RBD was codon-optimized and synthesized. After adding signal peptide and Kozak sequences to N terminus, the construct was cloned into the PTT5 plasmid vector via the *Hin* dIII and *Not* I restriction sites. The generated plasmid was validated by gene sequencing and then transfected into the CHO cells. After culture of 10–12 days, the supernatant was collected. Then, protein sample was purified by using ion-exchange chromatography and hydrophobic chromatography, followed by ultrafiltration. During purification, the samples from the eluted peaks were analyzed by SDS-PAGE and size exclusion-high-performance liquid chromatography (SEC-HPLC). The

homo-tri-RBD used in this study was stably expressed by CHO cells and purified following the similar processes described above.

## SPR assay

Surface plasmon resonance (SPR) assay was performed to quantify the binding avidity of the recombinant mos-tri-RBD to the receptor hACE2 using BIAcore 8 K (GE Healthcare) with NTA chips. Firstly, the His-tagged hACE2 protein was immobilized onto the chip surface. Then, the purified mos-tri-RBD protein sample was serially diluted in HBS-T buffer (HBS buffer and 0.05% Tween20). The diluted samples were injected at a flow rate of 30 μL/min for 120 s, and then the HBS-T buffer was flowed over the chip surface to facilitate dissociation of the bound protein for an additional 120 s. Subsequently, the sensor chip surface was regenerated by injecting 350 mM EDTA solution with a flow rate of 30 μL/min for 120 s. The SPR signal response was monitored as a function of time. BIAcoreTM Insight Evaluation software was used to analyze the experimental data. The binding kinetics between mos-tri-RBD and hACE2 was calculated using the software, and then the association and dissociation rate constants, that is, $k_a$ and $k_d$, as well as the apparent dissociation constant $K_D$ were obtained.

## Rat immunization

To test the immunogenicity of mos-tri-RBD, 10 Wistar rats with half male and half female (purchased from Beijing Vital River Laboratory Animal Technology Co., Ltd., China) were immunized intramuscularly by two doses, three weeks apart, of mos-tri-RBD (10 μg per dose) mixed with 300 μg aluminum hydroxide adjuvant. Another three groups of rats received two doses of homo-tri-RBD, BBIBP-CorV or adjuvant, respectively, with the same immunization regimen were used for comparison. On day 7 after full vaccination, sera from the immunized rats were collected.

To evaluate the immune efficacy of mos-tri-RBD as a booster dose, a total of 30 Wistar rats, half male and half female, were intramuscularly primed with a dose (4 μg/dose) of the inactivated vaccine BBIBP-CorV. After 3 weeks, 10 rats were boosted with 10 μg mos-tri-RBD mixed with 300 μg aluminum hydroxide adjuvant. The other 20 rats were boosted with a dose of homo-tri-RBD (10 μg antigen mixed with 300 μg aluminum hydroxide) or BBIBP-CorV (4 μg/dose), for comparison. Another 10 rats were vaccinated with two doses of adjuvant with the same immunization interval as control. The sera of all the immunized rats were collected on day 7 post-boosting immunization.

## ELISA

To verify the functionality of the recombinant protein mos-tri-RBD, enzyme-linked immunosorbent assay (ELISA) was employed to measure its binding activity with an RBD-specific monoclonal neutralizing antibody MM117 (purchased from Sino Biological Inc, China, Cat#40592-MM117) and the binding activities of the his-tagged monomeric prototype and Omicron RBDs with MM117 were also evaluated as control. Protein samples were prepared with the starting concentration of 1 μg/mL in carbonate buffer, and subjected to three-fold serial dilutions. The diluted samples were then pipetted into the wells of the ELISA plates with 100 μL per well followed by incubation at 2–8°C overnight. After removing the coating solution and washing the plate three times with PBS containing 0.05% Tween 20 (PBST), the remaining protein-binding sites were blocked by blocking buffer at 37 °C for 2 hr and the plate was again washed three times with PBST. The neutralizing antibody MM117 was prepared with the working concentration of 1 μg/mL, which was added to the plate with 100 μL per well and incubated at 37 °C for 1 hr. After washing three times with PBST, the plate was incubated with HRP-conjugated goat anti-mouse IgG antibody at 37 °C for 1 hr. Then, color reaction was developed with 50 μL tetramethylbenzidine (TMB) and 50 μL hydrogen peroxide solutions. After 5 min, color reactions were stopped with sulfuric acidic solution, and the optical density (OD) was measured both at 450 nm and 630 nm using the microplate reader. The difference in OD values at 450 nm and 630 nm, i.e. $OD_{450/630nm}$, was obtained to evaluate the specific binding activity of mos-tri-RBD with the neutralizing antibody MM117.

## Pseudo-virus neutralization assay

Neutralizing antibody levels in the sera from the immunized rats were detected by using pseudo-virus neutralization assay as described previously (*Liang et al., 2022*). In order to assess the cross-neutralization activities elicited by the vaccine candidate, we conducted neutralization assays against

10 pseudo-typed SARS-CoV-2 viruses, including prototype, Omicron (BA.1), Omicron (BA.2), Omicron (BA.3), Alpha, Beta, Delta, Gamma, Lambda and Mu strains. The pseudo-viruses were prepared by using the methods described in the previously published studies (*Zhang et al., 2022*; *Li et al., 2021*; *Zhang et al., 2021*; *Nie et al., 2020*). In short, the gene sequences of the S protein of SARS-CoV-2 variants were codon-optimized and synthesized, which were then cloned into pcDNA3.1 plasmid vector. The constructed plasmids encoding the S protein of SARS-CoV-2 variants were transfected into HEK293T cells, which were simultaneously infected with G*ΔG-VSV. After 24 hr, the culture supernatants were collected and filtered with 0.45 µm membrane filters to obtain the VSV-based pseudo-typed SARS-CoV-2 variants. The TICD$_{50}$ of the pseudo-viruses was determined by using Huh-7 cells.

To evaluate the pseudo-virus neutralization activities of the serum samples, the sera were firstly 1:40 diluted, followed by a fivefold serial dilution with the cell culture medium. The pseudo-typed SARS-CoV-2 strains were also diluted to the titer of $1.3 \times 10^4$ TCID$_{50}$ per mL. Then, 50 µL diluted serum was mixed with an equal volume of pseudo-virus in the well of the plates, and incubated at 37 °C and 5% CO$_2$ for 1 hr. Fifty µL culture medium mixed with 50 µL pseudo-virus was used as a control, and 100 µL per well medium without adding pseudo-virus served as a blank control. Subsequently, 100 µL trypsin-treated Huh-7 cells with the density of $2 \times 10^5$ per mL were added into the well of the plates, and incubated at 37 °C and 5% CO$_2$ for 20~24 hr. After that, the cells were lysed and the luminescence signals were detected by microplate luminometer using luciferase substrate. The neutralizing antibody titer was determined as the reciprocal of the serum dilution causing 50% reduction (ID50) in relative light units (RLUs), which was calculated using the Reed-Muench method. The ID50 for the serum below the limit of quantification (LOQ) was set to the quantification limit, that is, 40 in this assay.

## Live virus neutralization assay

All the viruses used in the live virus neutralization assay, including the prototype, Beta, Delta and Omicron strains, were obtained from the National Institute for Viral Disease Control and Prevention, Chinese Center for Disease Control and Prevention (China CDC), Beijing, China. The prototype (QD-01 strain) virus was isolated from Qingdao, and the Beta (GD84 strain) and Delta (GD96 strain) viruses were isolated from Guangdong. All these three viruses infect Vero cells well. For the Omicron virus, multiple strains isolated from different places, including Hong Kong, Shanghai, Tianjin and Changchun, were screened. It was found that not all the strains infect Vero cells well. After serial passages in Vero cells, the Omicron strain (BA.1.1, a sublineage of BA.1; No. NPRC 2.192100005) isolated from Shanghai was selected, which adapted to and propagated well in Vero cells. Typical cytopathic changes of Vero cells caused by Omicron infection can be observed after 5–7 days of culture. The genome of this Omicron BA.1.1 strain after passages was sequenced. *Appendix 1— figure 1* shows the gene sequence of the spike region of the screened Omicron BA.1.1 virus. The corresponding amino acid sequence of the spike protein was aligned with that of the prototype virus, as shown in *Appendix 1—figure 2*. The alignment result shows that there are 40 residue mutations, deletions or insertions in the spike region compared to that of the prototype virus. All these mutations are commonly occurred in BA.1.1 sub-lineage with >70% prevalence according to the analysis by the Lineage Comparison Tool of the outbreak.info web server (https://outbreak.info/compare-lineages). BLAST search against the sequences collected in GISAID showed that many Omicron BA.1.1 spike gene sequences with 100% identity to our sequence have been reported (https://www.epicov.org/epi3/frontend#2ccaab).

The live virus neutralization assay was performed in the BSL3 facility of the National Institute for Viral Disease Control and Prevention, China CDC, Beijing, China. The neutralization assay adopted in our study was based on the inhibition of the cytopathic effect (CPE). In the assay, serum samples were heat-inactivated at 56 °C for 30 min, and diluted 1:20 using cell culture medium 199 (M199). Then, the sera were two-fold serially diluted in 96-well plate with 50 µL per well. An equal volume of SARS-CoV-2 solution containing 100 TCID$_{50}$ of live virus was added to the well and mixed with the serum. The serum-virus mixed solution was cultured 2 hr in the incubator maintaining a consistent temperature of 37 °C and 5% carbon dioxide (CO$_2$). After incubation, Vero cell suspension, with a density of $(1.5–2) \times 10^5$ per mL, in the medium M199 that contains 5% FBS was added into the mixture with 100 µL per well. Both negative serum and positive reference serum (obtained from the National Institute for Food and Drug Control of China) were included in the plate as controls. Cell control (no virus and no tested serum) was also included. The titer of the virus was also titrated as a comparison in

the assay. Subsequently, the plates were incubated at 37 °C for 5–7 days in the incubator maintaining a consistent temperature of 37 °C and 5% $CO_2$. Then, the cellular changes caused by CPE were observed under the microscope, and the wells with cytopathic changes were recorded. Neutralizing antibody titer was reported as the reciprocal of the highest serum dilution that could inhibit 50% CPE (ID50), which was calculated by the Karber method. The ID50 for the serum below the LOQ of 20 in the assay was set to 20.

## Quantification and statistical analysis

One-way ANOVA with the LSD t-test method was used for the comparison of data from multiple groups, and Student's t-test was used for statistical analysis between two groups. $*p < 0.05$, $**p < 0.01$, $***p < 0.001$, $**** < 0.0001$. Details can be found in the figure legend.

## Acknowledgements

This work was supported by National Vaccine and Serum Institute (KTZC1900026C). Jing Z and QML were supported by National Vaccine and Serum Institute (KTZC1900026C). The funders had no role in study design, data collection and interpretation, or the decision to submit the work for publication.

## Additional information

### Competing interests

Jing Zhang, Zi Bo Han, Yu Liang, Xue Feng Zhang, Yu Qin Jin, Li Fang Du, Shuai Shao, Jun Wei Hou, Ze Hua Lei, Zhao Ming Liu, Ya Nan Hou, Ning Liu, Fu Jie Shen, Ji Guo Su, Qi Ming Li: is listed as an inventor of the pending patent application for the mos-tri-RBD vaccine (Application number: 202210083654.X). Hui Wang, Jin Zhang: is an employee of Beijing Institute of Biological Products Company Limited. The other authors declare that no competing interests exist.

### Funding

| Funder | Grant reference number | Author |
|---|---|---|
| National Vaccine and Serum Institute | KTZC1900026C | Jing Zhang Qi Ming Li |

The funders had no role in study design, data collection and interpretation, or the decision to submit the work for publication.

### Author contributions

Jing Zhang, Conceptualization, Supervision, Methodology, Writing – original draft; Zi Bo Han, Investigation, Visualization, Methodology, Writing – original draft; Yu Liang, Conceptualization, Investigation, Visualization, Methodology, Writing – original draft; Xue Feng Zhang, Yu Qin Jin, Li Fang Du, Jun Wei Hou, Ke Xu, Ya Nan Hou, Jin Juan Wu, Xin Yu Li, Xin Li, Investigation; Shuai Shao, Validation, Writing – original draft; Hui Wang, Wenwen Lei, Jin Zhang, Resources; Ze Hua Lei, Zhao Ming Liu, Ning Liu, Fu Jie Shen, Xiang Zheng, Validation; Wei Jin Huang, Resources, Writing – review and editing; Gui Zhen Wu, Resources, Supervision, Methodology, Writing – review and editing; Ji Guo Su, Conceptualization, Visualization, Methodology, Writing – original draft; Qi Ming Li, Conceptualization, Supervision, Methodology, Writing – review and editing

### Author ORCIDs

Ji Guo Su ⓘ http://orcid.org/0000-0003-0778-3477
Qi Ming Li ⓘ http://orcid.org/0000-0001-8284-7106

### Ethics

Animal experiments were approved by the Institutional Animal Care and Use Committee (IACUC) of the National Vaccine and Serum Institute (NVSI) (No. NVSI-RCD-JSDW-ER-2021238, NVSI-RCD-JSDW-ER-2022015) and conducted under the regulations for the administration of affairs concerning experimental animals of China (2017).

**Decision letter and Author response**
Decision letter https://doi.org/10.7554/eLife.78633.sa1
Author response https://doi.org/10.7554/eLife.78633.sa2

## Additional files

### Supplementary files
• Transparent reporting form

### Data availability
Figure 1—source data 1, Figure 1—source data 2, Figure 2—source data 1, Figure 3—source data 1 and Figure 4—source data 1 contain the numerical data used to generate the figures. The gene sequence of the spike region of the Omicron BA.1.1 virus used in the live virus neutralization assay is provided in Appendix 1—figure 1, and the residue mutations in the Omicron BA.1.1 spike region compared to that of the prototype virus are provided in Appendix 1—figure 2.

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

## Appendix 1

ATGTTTGTTTTTCTTGTTTTATTGCCACTAGTCTCTAGTCAGTGTGTTAATCTTACAACCAGAACTCAATTACCCCCTGCATACACTAATTCTT
TCACACGTGGTGTTTATTACCCTGACAAAGTTTTCAGATCCTCAGTTTTACATTCAACTCAGGACTTGTTCTTACCTTTCTTTTCCAATGTTA
CTTGGTTCCATGTTATCTCTGGGACCAATGGTACTAAGAGGTTTGATAACCCTGTCCTACCATTTAATGATGGTGTTTATTTTGCTTCCATTG
AGAAGTCTAACATAATAAGAGGCTGGATTTTTGGTACTACTTTAGATTCGAAGACCCAGTCCCTACTTATTGTTAATAACGCTACTAATGTT
GTTATTAAAGTCTGTGAATTTCAATTTTGTAATGATCCATTTTTGGACCACAAAAACAACAAAAGTTGGATGGAAAGTGAGTTCAGAGTTT
ATTCTAGTGCGAATAATTGCACTTTTGAATATGTCTCTCAGCCTTTTCTTATGGACCTTGAAGGAAAACAGGGTAATTTCAAAAATCTTAGG
GAATTTGTGTTTAAGAATATTGATGGTTATTTTAAAATATATTCTAAGCACACGCCTATTATAGTGCGTGAGCCAGAAGATCTCCCTCAGG
GTTTTTCGGCTTTAGAACCATTGGTAGATTTGCCAATAGGTATTAACATCACTAGGTTTCAAACTTTACTTGCTTTACATAGAAGTTATTTGA
CTCCTGGTGATTCTTCTTCAGGTTGGACAGCTGGTGCTGCAGCTTATTATGTGGGTTATCTTCAACCTAGGACTTTTCTATTAAAATATAAT
GAAAATGGAACCATTACAGATGCTGTAGACTGTGCACTTGACCCTCTCTCAGAAACAAAGTGTACGTTGAAATCCTTCACTGTAGAAAAA
GGAATCTATCAAACTTCTAACTTTAGAGTCCAACCAACAGAATCTATTGTTAGATTTCCTAATATTACAAACTTGTGCCCTTTTGATGAAGT
TTTTAACGCCACCAAATTTGCATCTGTTTATGCTTGGAACAGGAAGAGAATCAGCAACTGTGTTGCTGATTATTCTGTCCTATATAATCTCG
CACCATTTTTCACTTTTAAGTGTTATGGAGTGTCTCCTACTAAATTAAATGATCTCTGCTTTACTAATGTCTATGCAGATTCATTTGTAATTAG
AGGTGATGAAGTCAGACAAATCGCTCCAGGGCAAACTGGAAATATTGCTGATTATAATTATAAATTACCAGATGATTTTACAGGCTGCGT
TATAGCTTGGAATTCTAACAAGCTTGATTCTAAGGTTAGTGGTAATTATAATTACCTGTATAGATTGTTTAGGAAGTCTAATCTCAAACCTTT
TGAGAGAGATATTTCAACTGAAATCTATCAGGCCGGTAACAAACCTTGTAATGGTGTTGCAGGTTTTAATTGTTACTTTCCTTTACGATCAT
ATAGTTTCCGACCCACTTATGGTGTTGGTCACCAACCATACAGAGTAGTAGTACTTTCTTTTGAACTTCTACATGCACCAGCAACTGTTTGT
GGACCTAAAAAGTCTACTAATTTGGTTAAAAACAAATGTGTCAATTTCAACTTCAATGGTTTAAAAGGCACAGGTGTTCTTACTGAGTCTA
ACAAAAAGTTTCTGCCTTTCCAACAATTTGGCAGAGACATTGCTGACACTACTGATGCTGTCCGTGATCCACAGACACTTGAGATTCTTGA
CATTACACCATGTTCTTTTGGTGGTGTGTCAGTGTTATAACACCAGGAACAAATACTTCTAACCAGGTTGCTGTTCTTTATCAGGGTGTTAACT
GCACAGAAGTCCCTGTTGCTATTCATGCAGATCAACTTACTCCTACTTGGCGTGTTTATTCTACAGGTTCTAATGTTTTTCAAACACGTGCA
GGCTGTTTAATAGGGGCTGAATATGTCAACAACTCATATGAGTGTGACATACCCATTGGTGCAGGTATATGCGCTAGTTATCAGACTCAG
ACTAAGTCTCATCGGCGGGCACGTAGTGTAGCTAGTCAATCCATCATTGCCTACACTATGTCACTTGGTGCAGAAAATTCAGTTGCTTACT
CTAATAACTCTATTGCCATACCCACAAATTTTACTATTAGTGTTACCACAGAAATTCTACCAGTGTCTATGACCAAGACATCAGTAGATTGT
ACAATGTACATTTGTGGTGATTCAACTGAATGCAGCAATCTTTTGTTGCAATATGGCAGTTTTTGTACACAATTAAAACGTGCTTTAACTGG
AATAGCTGTTGAACAAGACAAAAACACCCAAGAAGTTTTTGCACAAGTCAAACAAATTTACAAAACACCACCAATTAAATATTTTGGTGG
TTTTAATTTTTCACAAATATTACCAGATCCATCAAAACCAAGCAAGAGGTCATTTATTGAAGATCTACTTTTCAACAAAGTGACACTTGCAG
ATGCTGGCTTCATCAAACAATATGGTGATTGCCTTGGTGATATTGCTGCTAGAGACCTCATTTGTGCACAAAAGTTTAAAGGCCTTACTGT
TTTGCCACCTTTGCTCACAGATGAAATGATTGCTCAATACACTTCTGCACTGTTAGCGGGTACAATCACTTCTGGTTGGACCTTTGGTGCA
GGTGCTGCATTACAAATACCATTTGCTATGCAAATGGCTTATAGGTTTAATGGTATTGGAGTTACACAGAATGTTCTCTATGAGAACCAAA
AATTGATTGCCAACCAATTTAATAGTGCTATTGGCAAAATTCAAGACTCACTTTCTTCCACAGCAAGTGCACTTGGAAAACTTCAAGATGT
GGTCAACCATAATGCACAAGCTTTAAACACGCTTGTTAAACAACTTAGCTCCAAATTTGGTGCAATTTCAAGTGTTTTAAATGATATCTTTT
CACGTCTTGACAAAGTTGAGGCTGAAGTGCAAATTGATAGGTTGATCACAGGCAGACTTCAAAGTTTGCAGACATATGTGACTCAACAAT
TAATTAGAGCTGCAGAAATCAGAGCTTCTGCTAATCTTGCTGCTACTAAAATGTCAGAGTGTGTACTTGGACAATCAAAAAGAGTTGATTT
TTGTGGAAAGGGCTATCATCTTATGTCCTTCCCTCAGTCAGCACCTCATGGTGTAGTCTTCTTGCATGTGACTTATGTCCCTGCACAAGAAA
AGAACTTCACAACTGCTCCTGCCATTTGTCATGATGGAAAAGCACACTTTCCTCGTGAAGGTGTCTTTGTTTCAAATGGCACACACTGGTT
TGTAACACAAAGGAATTTTTATGAACCACAAATCATTACTACAGACAACACATTTGTGTCTGGTAACTGTGATGTTGTAATAGGAATTGTC
AACAACACAGTTTATGATCCTTTGCAACCTGAATTAGATTCATTCAAGGAGGAGTTAGATAAATATTTTAAGAATCATACATCACCAGATG
TTGATTTAGGTGACATCTCTGGCATTAATGCTTCAGTTGTAAACATTCAAAAAGAAATTGACCGCCTCAATGAGGTTGCCAAGAATTTAAA
TGAATCTCTCATCGATCTCCAAGAACTTGGAAAGTATGAGCAGTATATAAAATGGCCATGGTACATTTGGCTAGGTTTTATAGCTGGCTTG
ATTGCCATAGTAATGGTGACAATTATGCTTTGCTGTATGACCAGTTGCTGTAGTTGTCTCAAGGGCTGTTGTTCTTGTGGATCCTGCTGCAA
ATTTGATGAAGACGACTCTGAGCCAGTGCTCAAAGGAGTCAAATTACATTACACATAA

**Appendix 1—figure 1.** The gene sequence of the spike region of the Omicron BA.1.1 virus used in the live virus neutralization assay.

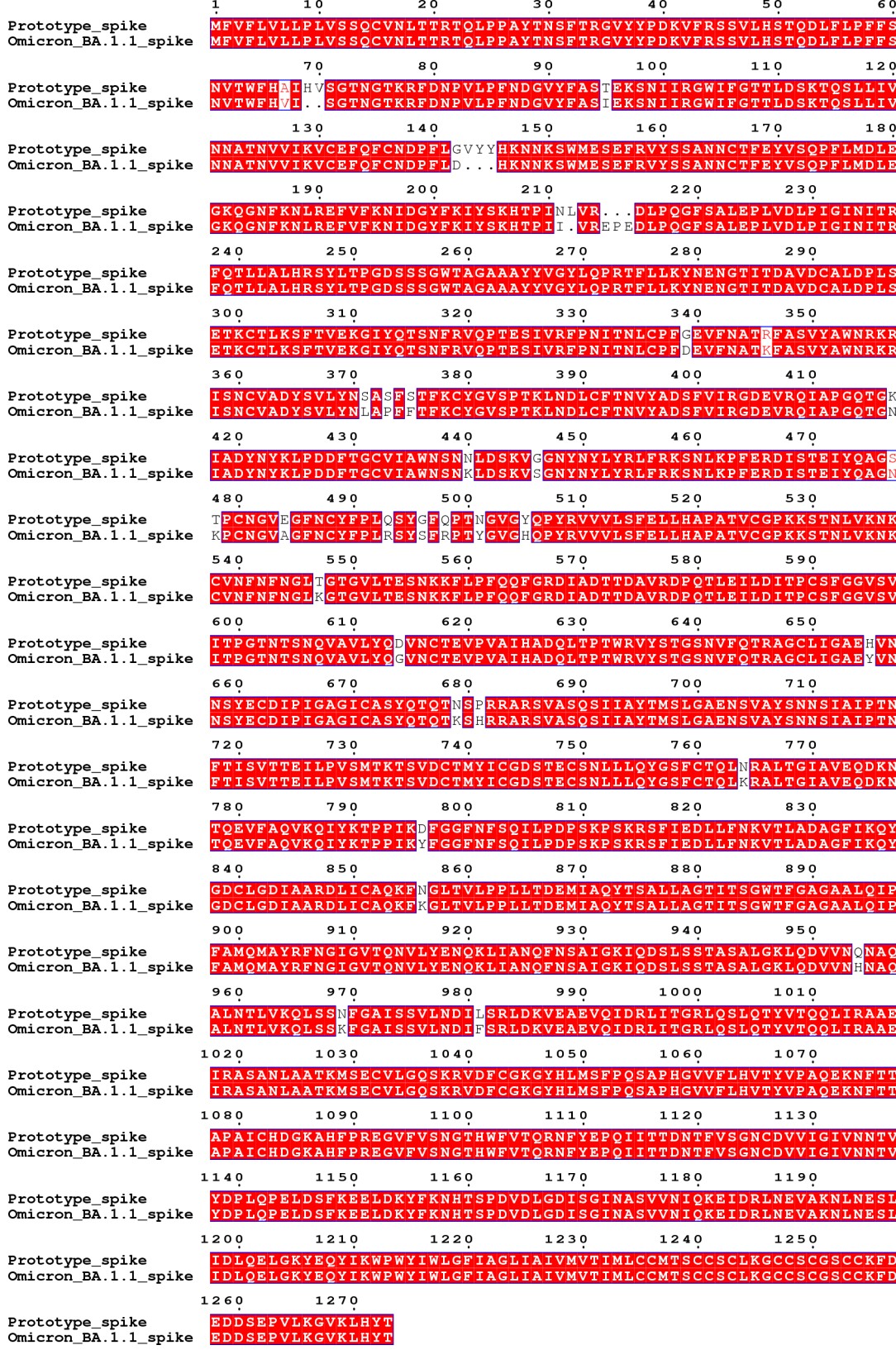

**Appendix 1—figure 2.** The amino acid sequence of the spike region of the Omicron BA.1.1 virus used in the live virus neutralization assay aligned with that of the prototype virus. The sequence alignment result shows that there are 40 residue mutations, deletions or insertions in the spike region compared to that of the prototype virus,

*Appendix 1—figure 2 continued on next page*

*Appendix 1—figure 2 continued*
including A67V, del69-70, T95I, G142D, del143-145, N211I, del212, insert EPE, G339D, R346K, S371L, S373P, S375F, K417N, N440K, G446S, S477N, T478K, E484A, Q493R, G496S, Q498R, N501Y, Y505H, T547K, D614G, H655Y, N679K, P681H, N764K, D796Y, N856K, Q954H, N969K and L981F.

