## [Editor Report]

In this work, the authors test, in an animal model, a vaccine booster incorporating three linked SARS-CoV-2 spike receptor binding domain sub-units containing mutations from Omicron sub-variant BA.1 as well as other variants. They demonstrate that this is more effective at boosting neutralizing immunity against Omicron sub-variants and other variants including Beta and Delta than the same vaccine design but incorporating ancestral SARS-CoV-2 spike. While vaccine manufacturers are currently racing to make boosters based on Omicron sub-variant sequences, the approach presented in this paper, which combines mutations in addition to those found on individual Omicron sub-variant sequences, may offer another perspective on how to boost previous vaccine immunity to tackle emerging variants.

---

## [Decision Letter]

**Decision letter after peer review:**

[Editors’ note: the authors submitted for reconsideration following the decision after peer review. What follows is the decision letter after the first round of review.]

Thank you for submitting the paper "A mosaic-type trimeric RBD-based COVID-19 vaccine candidate induces potent neutralization against Omicron and other SARS-CoV-2 variants" for consideration by *eLife*. Your article has been reviewed by 2 peer reviewers, one of whom is a member of our Board of Reviewing Editors, and the evaluation has been overseen by a Senior Editor. The following individual involved in review of your submission have agreed to reveal their identity: Pragati Agnihotri (Reviewer #2).

Comments to the Authors:

We are sorry to say that, after consultation with the reviewers, we have decided that this work will not be considered further for publication by *eLife*.

While there was enthusiasm for the work, since the results have implications for designing vaccines and therefore people's health, the experiments should be clearly described.

The methodology in this paper was not clear to one of the reviewers, including fundamental aspects such as the number of animals used per experiment, the limit of detection/quantification, and how the live virus neutralization assay was done. These technical points are key to evaluating the reproducibility of the results. Information such as annotated images of the raw data in the live virus neutralization assay may be provided to help the reviewers evaluate the work, and the results shown in the figure panels should be fully explained in the figure legends and text. Consideration may be given to a rebuttal if these technical points are resolved.

*Reviewer #1 (Recommendations for the authors):*

More broadly acting SARS-CoV-2 vaccines are required and the approach of the authors to use a mosaic vaccine is reasonable. The major drawback of the work is that it is unclear exactly how the work was done, making it difficult to evaluate.

The methodology behind the live virus neutralization assay, a key assay in the paper, is unclear to me. In the methods, the authors state that "The serum virus mixed solution was incubated at 37 ℃ for 2 h. After incubation, Vero cell suspension with a density of 2×105 per mL was added into the mixture. Cell-only and virus-only wells were also included in the plate as controls. The plates were incubated at 37 ℃ for 5 to 7 days." What exactly is measured here, plaques or foci? How is the fact that Omicron does not infect Vero cells well dealt with? The incubation time of 5 to 7 days is extremely long and would not work for most assays.

Similarly, it is unclear how many animals are used per experiment, although by counting the number of points it seems to be 9 or 10 per group. What the limit of quantification/detection for the assay is unclear, although it may be a 1:10 dilution judging that at the adjuvant only control as the points are at that value in Figures 2-4. The meaning of the different lines in Figure 1D is not well explained.

Given that the approach described has implications for designing vaccines and so has a direct bearing on population health, the methodology should be more rigorously described.

*Reviewer #2 (Recommendations for the authors):*

The manuscript deserved to be published in eLife.

[Editors’ note: further revisions were suggested prior to acceptance, as described below.]

Thank you for resubmitting your work entitled "A mosaic-type trimeric RBD-based COVID-19 vaccine candidate induces potent neutralization against Omicron and other SARS-CoV-2 variants" for further consideration by *eLife*. Your revised article has been evaluated by Betty Diamond (Senior Editor) and a Reviewing Editor.

The manuscript has been improved but there are some remaining issues that need to be addressed, as outlined below:

1) The authors clarified how the live virus neutralization results were obtained in the rebuttal letter but this explanation should also be included in the Methods section and more details should be added.

Specifically, as the authors acknowledge, Omicron BA.1 infection of Vero cells is generally inefficient and may not be very cytopathic, yet the authors quantified the cytopathic effect. Vero infection and the cytopathic effect were achieved with a Vero passaged BA.1.1, which likely accumulated mutations in vitro which allowed for this. While mutations outside of the spike may not strongly affect neutralization, the authors should present the sequence of the spike region of the BA1.1 isolate they used to show that no spike mutations were introduced.

It would also be very helpful if the sequences of viruses used in this work were to be deposited in a repository such as GISAID and the accessions listed.

2) In Figure 2 and Figure 4, the authors set titer values below the limit of quantification (which was 20) to 10. This seems arbitrary. Values below 20 should be set to 20, extrapolated using the fit, or listed as below LOQ.

---

## [Author Response]

[Editors’ note: The authors appealed the original decision. What follows is the authors’ response to the first round of review.]

While there was enthusiasm for the work, since the results have implications for designing vaccines and therefore people's health, the experiments should be clearly described.

The methodology in this paper was not clear to one of the reviewers, including fundamental aspects such as the number of animals used per experiment, the limit of detection/quantification, and how the live virus neutralization assay was done. These technical points are key to evaluating the reproducibility of the results. Information such as annotated images of the raw data in the live virus neutralization assay may be provided to help the reviewers evaluate the work, and the results shown in the figure panels should be fully explained in the figure legends and text. Consideration may be given to a rebuttal if these technical points are resolved.

Thank you for your insightful and constructive comments on our manuscript. We would also appreciate you allowing us to clarify the key technical points raised by you and the reviewer.

The Omicron virus used in our study was isolated and cultured by the National Institute for Viral Disease Control and Prevention, Chinese Center for Disease Control and Prevention (China CDC). Multiple Omicron strains isolated from different places, including Hong Kong, Shanghai, Tianjin and Changchun, were screened. As pointed out by the reviewer, not all the strains infect Vero cells well. After serial passages in Vero cells, the Omicron strain (BA.1.1, a sublineage of BA.1; No. NPRC 2.192100005) isolated from Shanghai was selected, which adapted to and propagated well in Vero cells. Typical cytopathic changes of Vero cells caused by Omicron infection can be observed after 5-7 days of incubation. It was found that after infection with Omicron virus, the Vero cells lose their normal shape, becoming round or shrinking. Some cells start to detach and are afloat. The isolation and selection of Omicron virus were carried out by the National Institute for Viral Disease Control and Prevention, China CDC.

The live-virus neutralization assay adopted in our study was based on inhibition of cytopathic effect (CPE). The detailed procedure of the assay was as follows. Serum samples were heat-inactivated at 56 °C for 30 min, and diluted 1:20 using cell culture medium 199 (M199). Then, the sera were two-fold serially diluted in 96-well plate with 50 μL per well. An equal volume of SARS-CoV-2 solution containing 100 TCID_50_ of live virus was added to the well and mixed with the serum. The serum-virus mixed solution was cultured 2 h in the incubator maintaining a consistent temperature of 37°C and 5% carbon dioxide (CO_2_). After incubation, Vero cell suspension, with a density of (1.5-2)×10^5^ per mL, in the medium M199 that contains 5% fetal bovine serum was added into the mixture with 100 μL per well. Both negative serum and positive reference serum (obtained from the National Institute for Food and Drug Control of China) were included in the plate as controls. Cell control (no virus and no tested serum) was also included. The titer of the virus was also titrated as a comparison in the assay. Subsequently, the plates were incubated for 5 to 7 days in the incubator maintaining a consistent temperature of 37°C and 5% CO_2_. Then, the cellular changes caused by CPE were observed under the microscope, and the wells with cytopathic changes were recorded. Neutralizing antibody titer was reported as the reciprocal of the highest serum dilution that could inhibit 50% CPE, which was calculated by the Karber method. The titer for the serum below the detection limit of 20 in the assay was set to 10. The procedure of the live-virus neutralization assay has been described in detail in the revised manuscript. (Please see pages 30-31 highlighted in yellow).

Unfortunately, due to the limitation of the experimental equipment and the consideration of biosafety, no images for the cytopathic effect of Vero cells in the live-virus neutralizing assay were obtained from the BSL3 facility of the National Institute for Viral Disease Control and Prevention, China CDC.

Our live-virus neutralizing antibody detection method has been validated carefully and used to evaluate the immunogenicity of the homo-tri-RBD vaccine developed by our group in a phase 2 trial [Ref: Kaabi NA, et al. (2022) Immunogenicity and safety of NVSI-06-07 as a heterologous booster after priming with BBIBP-CorV: a phase 2 trial. Signal Transduction and Targeted Therapy 7:171.] The specificity, repeatability, intermediate precision, linearity, relative accuracy and virus titer titration have been verified, respectively. The validation process and results met all acceptance criteria defined in the validation protocol, and a standard operating procedure (SOP) has been formulated. During the validation of the detection method, the WHO international reference materials were tested by using our method, and the detected neutralizing antibody titer was 320, which was consistent with the results reported by other studies [Ref: Li J, et al. (2022) Heterologous AD5-nCOV plus CoronaVac versus homologous CoronaVac vaccination: a randomized phase 4 trial. Nature Medicine 28:401-409.]. In addition, 18 human convalescent serum samples were tested by using our method, and the detected GMT value was 56.3, which agreed well with previously reported results obtained by other groups[Ref: Yang S, et al. (2021) Safety and immunogenicity of a recombinant tandem-repeat dimeric RBD-based protein subunit vaccine (ZF2001) against COVID-19 in adults: two randomised, double-blind, placebo-controlled, phase 1 and 2 trials. Lancet Infectious Diseases 21:1107-1119.]. Therefore, the method has been determined to be reliable and effective for the detection of neutralizing antibody titers in serum samples.

The number of animals used in each experiment, the detection limit of the pseudo- and live-virus neutralization assays, and the meaning of different curves in Figure 1D have been provided in the corresponding figure legends and the text, as suggested by the editor and the first reviewer.

Reviewer #1 (Recommendations for the authors):More broadly acting SARS-CoV-2 vaccines are required and the approach of the authors to use a mosaic vaccine is reasonable. The major drawback of the work is that it is unclear exactly how the work was done, making it difficult to evaluate.The methodology behind the live virus neutralization assay, a key assay in the paper, is unclear to me. In the methods, the authors state that "The serum virus mixed solution was incubated at 37 °C for 2 h. After incubation, Vero cell suspension with a density of 2×105 per mL was added into the mixture. Cell-only and virus-only wells were also included in the plate as controls. The plates were incubated at 37 °C for 5 to 7 days." What exactly is measured here, plaques or foci? How is the fact that Omicron does not infect Vero cells well dealt with? The incubation time of 5 to 7 days is extremely long and would not work for most assays.Similarly, it is unclear how many animals are used per experiment, although by counting the number of points it seems to be 9 or 10 per group. What the limit of quantification/detection for the assay is unclear, although it may be a 1:10 dilution judging that at the adjuvant only control as the points are at that value in Figures 2-4. The meaning of the different lines in Figure 1D is not well explained.Given that the approach described has implications for designing vaccines and so has a direct bearing on population health, the methodology should be more rigorously described.

Thank you for your critical and insightful comments on our manuscript. We understand your concerns which are caused by the unclear description in the manuscript. We would appreciate it if you could allow us to clarify the issues you raised.

The Omicron virus used in our study was isolated and cultured by the National Institute for Viral Disease Control and Prevention, Chinese Center for Disease Control and Prevention (China CDC). Multiple Omicron strains isolated from different places, including Hong Kong, Shanghai, Tianjin and Changchun, were screened. As pointed out by the reviewer, not all the strains infect Vero cells well. After serial passages in Vero cells, the Omicron strain (BA.1.1, a sublineage of BA.1; No. NPRC 2.192100005) isolated from Shanghai was selected, which adapted to and propagated well in Vero cells. Typical cytopathic changes of Vero cells caused by Omicron infection can be observed after 5-7 days of incubation. It was found that after infection with Omicron virus, the Vero cells lose their normal shape, becoming round or shrinking. Some cells start to detach and are afloat. The isolation and selection of Omicron virus were carried out by the National Institute for Viral Disease Control and Prevention, China CDC.

The live-virus neutralization assay adopted in our study was based on inhibition of cytopathic effect (CPE). The detailed procedure of the assay was as follows. Serum samples were heat-inactivated at 56 °C for 30 min, and diluted 1:20 using cell culture medium 199 (M199). Then, the sera were two-fold serially diluted in 96-well plate with 50 μL per well. An equal volume of SARS-CoV-2 solution containing 100 TCID_50_ of live virus was added to the well and mixed with the serum. The serum-virus mixed solution was cultured 2 h in the incubator maintaining a consistent temperature of 37°C and 5% carbon dioxide (CO_2_). After incubation, Vero cell suspension, with a density of (1.5-2)×10^5^ per mL, in the medium M199 that contains 5% fetal bovine serum was added into the mixture with 100 μL per well. Both negative serum and positive reference serum (obtained from the National Institute for Food and Drug Control of China) were included in the plate as controls. Cell control (no virus and no tested serum) was also included. The titer of the virus was also titrated as a comparison in the assay. Subsequently, the plates were incubated for 5 to 7 days in the incubator maintaining a consistent temperature of 37°C and 5% CO_2_. Then, the cellular changes caused by CPE were observed under the microscope, and the wells with cytopathic changes were recorded. Neutralizing antibody titer was reported as the reciprocal of the highest serum dilution that could inhibit 50% CPE, which was calculated by the Karber method. The titer for the serum below the detection limit of 20 in the assay was set to 10. The procedure of the live-virus neutralization assay has been described in detail in the revised manuscript. (Please see pages 30-31).

Unfortunately, due to the limitation of the experimental equipment and the consideration of biosafety, no images for the cytopathic effect of Vero cells in the live-virus neutralizing assay were obtained from the BSL3 facility of the National Institute for Viral Disease Control and Prevention, China CDC.

Our live-virus neutralizing antibody detection method has been validated carefully and used to evaluate the immunogenicity of the homo-tri-RBD vaccine developed by our group in a phase 2 trial [Ref: Kaabi NA, et al. (2022) Immunogenicity and safety of NVSI-06-07 as a heterologous booster after priming with BBIBP-CorV: a phase 2 trial. *Signal Transduction and Targeted Therapy* 7:171.] The specificity, repeatability, intermediate precision, linearity, relative accuracy and virus titer titration have been verified, respectively. The validation process and results met all acceptance criteria defined in the validation protocol, and a standard operating procedure (SOP) has been formulated. During the validation of the detection method, the WHO international reference materials were tested by using our method, and the detected neutralizing antibody titer was 320, which was consistent with the results reported by other studies [Ref: Li J, et al. (2022) Heterologous AD5-nCOV plus CoronaVac versus homologous CoronaVac vaccination: a randomized phase 4 trial. *Nature Medicine* 28:401-409.]. In addition, 18 human convalescent serum samples were tested by using our method, and the detected GMT value was 56.3, which agreed well with previously reported results obtained by other groups [Ref: Yang S, et al. (2021) Safety and immunogenicity of a recombinant tandem-repeat dimeric RBD-based protein subunit vaccine (ZF2001) against COVID-19 in adults: two randomised, double-blind, placebo-controlled, phase 1 and 2 trials. *Lancet Infectious Diseases* 21:1107-1119.]. Therefore, the method has been determined to be reliable and effective for the detection of neutralizing antibody titers in serum samples.

In the live-virus neutralization assay, the Vero cells were cultured in Medium 199 containing 5% fetal bovine serum, which are growing well during the incubation time of 5-7 days.

For all the animal experiments reported in the manuscript, 10 Wistar rats, half male and half female, were used in each group. The exact number of animals per group was provided in the corresponding figure legends in the revised manuscript. (Please see pages 12 and14).

In our study, the detection limit of the live-virus neutralization assay was 20, and the titers below the limit of detection were set to 10. For the pseudo-virus neutralization assay, the detection limit was 40, and the titers below the limit of detection were set to 20. This information has been provided in the revised manuscript. (Please see pages 12, 14, 18 and 31).

The different curves in Figure 1D represent different concentrations of analyte in the SPR experiment, and the color and black curves display the original and fitted data, respectively. The meaning of these different curves has been added in the figure legend in the revised paper. (Please see page 9).

[Editors’ note: further revisions were suggested prior to acceptance, as described below.]

The manuscript has been improved but there are some remaining issues that need to be addressed, as outlined below:1) The authors clarified how the live virus neutralization results were obtained in the rebuttal letter but this explanation should also be included in the Methods section and more details should be added.Specifically, as the authors acknowledge, Omicron BA.1 infection of Vero cells is generally inefficient and may not be very cytopathic, yet the authors quantified the cytopathic effect. Vero infection and the cytopathic effect were achieved with a Vero passaged BA.1.1, which likely accumulated mutations in vitro which allowed for this. While mutations outside of the spike may not strongly affect neutralization, the authors should present the sequence of the spike region of the BA1.1 isolate they used to show that no spike mutations were introduced.It would also be very helpful if the sequences of viruses used in this work were to be deposited in a repository such as GISAID and the accessions listed.

Thank you for your insightful comments and suggestions on our manuscript. According to your suggestions, the screening and culture of Omicron virus used in the live virus neutralization assay were added in the Methods section of the revised manuscript. (Please see details on pages 31-32). Moreover, the sequence of the spike region of the isolated Omicron BA.1.1 virus was provided in the Appendix 1 in the revised manuscript. The sequencing result showed that there were 40 residue mutations, deletions or insertions in the spike region compared to the prototype virus. All these mutations commonly occurred in BA.1.1 sub-lineage with >70% prevalence, according to the analysis by the Lineage Comparison Tool of the outbreak.info web server [Ref: https://outbreak.info/compare-lineages]. BLAST search against the sequences collected in GISAID showed that many Omicron BA.1.1 spike gene sequences with 100% identity to our sequence have been reported [Ref: https://www.epicov.org/epi3/frontend#2ccaab]. All the above discussions were added to the revised paper. (Please see pages 31-32).

The genome sequencing of the isolated Omicron BA.1.1 virus was performed by Dr. Wenwen Lei from the National Institute for Viral Disease Control and Prevention, Chinese Center for Disease Control and Prevention (China CDC). Therefore, Dr. Wenwen Lei was added as an author in the revised paper, and all the other authors agree to the alteration of the authorship.

We appreciate your help and would consider your advice to upload the sequence to a repository. However, the decision can only be taken after close consultation with all related contributors to the isolation and culture of the viruses.

2) In Figure 2 and Figure 4, the authors set titer values below the limit of quantification (which was 20) to 10. This seems arbitrary. Values below 20 should be set to 20, extrapolated using the fit, or listed as below LOQ.

Thank you for your valuable suggestion. According to your suggestion, the ID50 titers below the limit of quantification were set to the quantification limits, i.e., 40 for the pseudo-virus neutralization assay and 20 for the live virus neutralization assay, in the revised manuscript. Then, all the GMT values and the related statistical analyses were re-calculated, and Figure 2, Figure 3 and Figure 4 were redrawn accordingly. (Please see details on pages 13, 15, 18, 19, 31 and 32).